# *wnt16* regulates spine and muscle morphogenesis through parallel signals from notochord and dermomyotome

Claire J. Watson[1,2], W. Joyce Tang[1,2], Maria F. Rojas[1,2], Imke A. K. Fiedler[3], Ernesto Morfin Montes de Oca[1,2], Andrea R. Cronrath[1,2], Lulu K. Callies[1,2], Avery Angell Swearer[1,2], Ali R. Ahmed[1,2], Visali Sethuraman[1,2], Sumaya Addish[1,2], Gist H. Farr, III[4], Arianna Ericka Gómez[1,2], Jyoti Rai[1,2], Adrian T. Monstad-Rios[1,2], Edith M. Gardiner[1,2], David Karasik[5], Lisa Maves[4,6], Bjorn Busse[3], Yi-Hsiang Hsu[5,7,8], Ronald Young Kwon[1,2]*

**1** Department of Orthopaedics and Sports Medicine, University of Washington School of Medicine, Seattle, Washington, United States of America, **2** Insitute for Stem Cell and Regenerative Medicines, University of Washington, Seattle Washington, United States of America, **3** Department of Osteology and Biomechanics, University Medical Center Hamburg-Eppendorf, Hamburg, Germany, **4** Center for Developmental Biology and Regenerative Medicine, Seattle Children's Research Institute, Seattle, Washington, United States of America, **5** Institute for Aging Research, Hebrew SeniorLife, Boston, Massachusetts, United States of America, **6** Department of Pediatrics, Division of Cardiology, University of Washington School of Medicine, Seattle, Washington, United States of America, **7** Department of Medicine, Harvard Medical School, Boston, Massachusetts, United States of America, **8** Broad Institute of Harvard and Massachusetts Institute of Technology, Boston, Massachusetts, United States of America

* ronkwon@uw.edu

**Data Availability Statement:** The MATLAB script used for lean tissue analysis is provided in Supplementary Data. MicroCT data for *wnt16*

## Abstract

Bone and muscle are coupled through developmental, mechanical, paracrine, and autocrine signals. Genetic variants at the *CPED1-WNT16* locus are dually associated with bone- and muscle-related traits. While *Wnt16* is necessary for bone mass and strength, this fails to explain pleiotropy at this locus. Here, we show *wnt16* is required for spine and muscle morphogenesis in zebrafish. In embryos, *wnt16* is expressed in dermomyotome and developing notochord, and contributes to larval myotome morphology and notochord elongation. Later, *wnt16* is expressed at the ventral midline of the notochord sheath, and contributes to spine mineralization and osteoblast recruitment. Morphological changes in *wnt16* mutant larvae are mirrored in adults, indicating that *wnt16* impacts bone and muscle morphology throughout the lifespan. Finally, we show that *wnt16* is a gene of major effect on lean mass at the *CPED1-WNT16* locus. Our findings indicate that Wnt16 is secreted in structures adjacent to developing bone (notochord) and muscle (dermomyotome) where it affects the morphogenesis of each tissue, thereby rendering *wnt16* expression into dual effects on bone and muscle morphology. This work expands our understanding of *wnt16* in musculoskeletal development and supports the potential for variants to act through *WNT16* to influence bone and muscle via parallel morphogenetic processes.

mutants and other lines can be viewed on our interactive FishCuT Phenome Viewer (https://msblgroup.shinyapps.io/phenoviewer/). The somite sc-RNA-seq analysis can be viewed on our interactive webtool at https://msblgroup.shinyapps.io/somite/. Additional data sets can be accessed at https://osf.io/q75zp/.

**Funding:** Research reported in this publication was supported by the National Institute of Arthritis and Musculoskeletal and Skin Diseases of the National Institutes of Health under Award Numbers AR074417 (R.Y.K.) and AR072199 (Y.-H.H.). The content is solely the responsibility of the authors and does not necessarily represent the official views of the National Institutes of Health. The authors would also like to acknowledge support from UW Royalty Research Fund Grant A139347 (R.Y.K.), a Seed Grant from the University of Washington Department of Orthopaedics and Sports Medicine (R.Y.K.), an Innovation Pilot Award from the Institute for Stem Cell and Regenerative Medicine (C.J.W.), and scholarships from the Mary Gates Endowment (A.R.A. and V.S.). The funders had no role in study design, data collection and analysis, decision to publish, or preparation of the manuscript.

**Competing interests:** The authors have declared that no competing interests exist.

## Author summary

In humans, genetic variants (DNA sequences that vary amongst individuals) have been identified that appear to influence two tissues, bone and skeletal muscle. However, how single genes and genetic variants exert dual influence on both tissues is not well understood. In this study, we found that the *wnt16* gene is necessary for specifying the size and shape of both muscle and bone during development in zebrafish. We also disentangled how *wnt16* affects both tissues: distinct cellular populations adjacent to muscle and bone secrete Wnt16, where it acts as a signal guiding the size and shape of each tissue. This is important because in humans, genetic variants near the *WNT16* gene have effects on both bone- and muscle-related traits. This study expands our understanding of the role of *WNT16* in bone and muscle development, and helps to explain how genetic variants near *WNT16* affect traits for both tissues. Moreover, WNT16 is actively being explored as a target for osteoporosis therapies, thus our study could have implications with regard to the potential of targeting WNT16 to treat bone and muscle simultaneously.

## Introduction

Bone and muscle are coupled through mechanical, developmental, paracrine, and autocrine signals [1]. Following organogenesis, both tissues acquire peak mass and then decline in size, usually in concert [2]. With advanced age, osteoporosis, a disease of bone fragility, and sarcopenia, a condition of reduced muscle mass and strength, frequently manifest in the same individual—a condition termed osteosarcopenia [3]. This is considered a "hazardous duet" [4], since sarcopenia increases susceptibility for falls, amplifying the risk of fracture in fragile osteoporotic bone. Genome-wide association studies (GWAS) have identified genetic variants with dual effects on bone- and muscle-related traits [5–8], indicating that bone and muscle mass have a shared genetic component. Pleiotropy may arise through variants having direct biological effects on each trait, or, through an effect on one trait which affects the other. Disentangling this could identify biological mechanisms underlying the coupling between bone and muscle, and lead to new approaches to treat both tissues simultaneously.

Chromosome region 7q31.31, also known as the *CPED1-WNT16* locus, harbors genetic variants with dual effects on bone- and muscle-related traits. Medina-Gomez et al. showed that genetic variants at the *CPED1-WNT16* locus are associated with pleiotropic effects on bone mineral density (BMD) and total body lean tissue mass, the latter of which is a clinical correlate of skeletal muscle mass [5]. These pleiotropic variants were identified in a pediatric population, highlighting their function early in life. Prior studies indicate a critical role of 7q31.31 and *WNT16* in influencing BMD. Human genetic studies have shown that variants in the region are associated with BMD and fracture risk [9–11]. *In vivo*, *Wnt16* knockout mice exhibit reduced cortical bone mass and strength, with these abnormalities phenocopied in mice with osteoblast-specific knockout of *Wnt16* [11, 12]. Moreover, WNT16 suppresses osteoclastogenesis *in vitro* [12]. Zebrafish with mutations in *wnt16* exhibit decreased bone mineral density [13] and bone fragility in fin rays [14]. Taken together, functional studies of *Wnt16* have shown it is necessary for bone mass and strength, however, effects on muscle have yet to be observed. Thus, our current understanding of the biology of *WNT16* fails to explain pleiotropy at the *CPED1-WNT16* locus.

It has been hypothesized that variants underlying bone and muscle pleiotropy may act during embryonic and fetal development, which is when bone and muscle undergo organogenesis through a highly interconnected gene network [5, 7]. In vertebrates, skeletal muscle and the

axial bone arise from somites, blocks of paraxial mesoderm that form during embryonic development. As the somite matures, it is divided into two major compartments: the sclerotome, which gives rise to vertebrae and ribs, and the dermomyotome, which gives rise to skeletal muscle and dermis [15, 16]. In zebrafish, the sclerotome and notochord work in concert to form the spine. Specifically, during spine development, notochord sheath cells pattern and mineralize the notochord sheath to form segmented mineralized domains termed chordacentra [17, 18]. Osteoblastic cells derived from sclerotome are then recruited to the mineralized domains of the notochord sheath, where they contribute to the formation of mature centra [18]. In regard to dermomyotome in zebrafish, the anterior portion of developing somites is enriched with *pax3/7*-expressing myogenic precursors [19]. At the conclusion of somite development, *pax3/7* is enriched within a single layer of cells superficial to the myotome termed the external cell layer, which is functionally equivalent to the amniote dermomyotome [20]. A previous study investigating hematopoietic stem cell development [21] showed that *wnt16* was enriched in developing somites in zebrafish. However, the study did not establish which of the somitic subdivisions (e.g., sclerotome, myotome, etc.) it was expressed in, nor its function in musculoskeletal development.

In this study, we examine the contribution of *wnt16* to spine and muscle organogenesis, and its potential relationship to pleiotropy at the *CPED1-WNT16* locus. For this we took advantage of the genetic and optical attributes of zebrafish [22, 23], and used a combination of microCT-based phenomics [24], single cell analysis, and CRISPR-based gene editing. The *CPED1-WNT16* locus comprises two variants independently associated with BMD (i.e. significant even after genetic linkage is accounted for): one near *WNT16* (lead SNP: rs3801387), and the other near *CPED1* (lead SNP: rs13245690). Because there is evidence these variants act on different regulatory elements [25], it has been hypothesized that the *CPED1-WNT16* locus comprises two independent signals that affect different genes. As such, we also examined whether other genes at the *CPED1-WNT16* locus are necessary for lean mass and morphology in zebrafish. For this, we employed methods for rapid reverse genetic screening recently described by our lab [26].

Here, we show a dual requirement of *wnt16* for spine and muscle morphogenesis. Moreover, we disentangle how *wnt16* affects both tissues: Wnt16 is secreted in notochord and dermomyotome, structures adjacent to developing bone and muscle, respectively, where it renders altered *wnt16* expression into effects on musculoskeletal form.

## Results

### Somitic *wnt16*+ cells are dermomyotome-like

To determine the identity of somitic *wnt16*+ cells, we performed single cell RNA-sequencing (scRNA-seq) analysis during zebrafish embryonic development. Using a published scRNA-seq atlas of zebrafish embryonic development [27], we subset cells from the somite cluster and performed subclustering and differential gene expression analyses (Fig 1A and 1A'). Using published marker genes, we defined 7 out of 10 of these subclusters as the external cell layer (dermomyotome-like), sclerotome, and differentiating muscle [19, 28]. Differential gene expression analysis revealed that *wnt16* was amongst the top 10 differentially expressed genes (9[th] most by p-value) in the dermomyotome-like 1 cluster (Fig 1B). Other top differentially expressed genes for the dermomyotome-like 1 cluster included *pax3a* and *pax7a*, two notable markers of early muscle specification, along with *emp2*, *rprmb*, *cep131*, *pleca*, *comp*, NC-00233.4, and *aldh1a2*. Notably, expression of *wnt16* was mostly absent in clusters identified as sclerotome (Fig 1B), a source of vertebral osteoblast precursors.

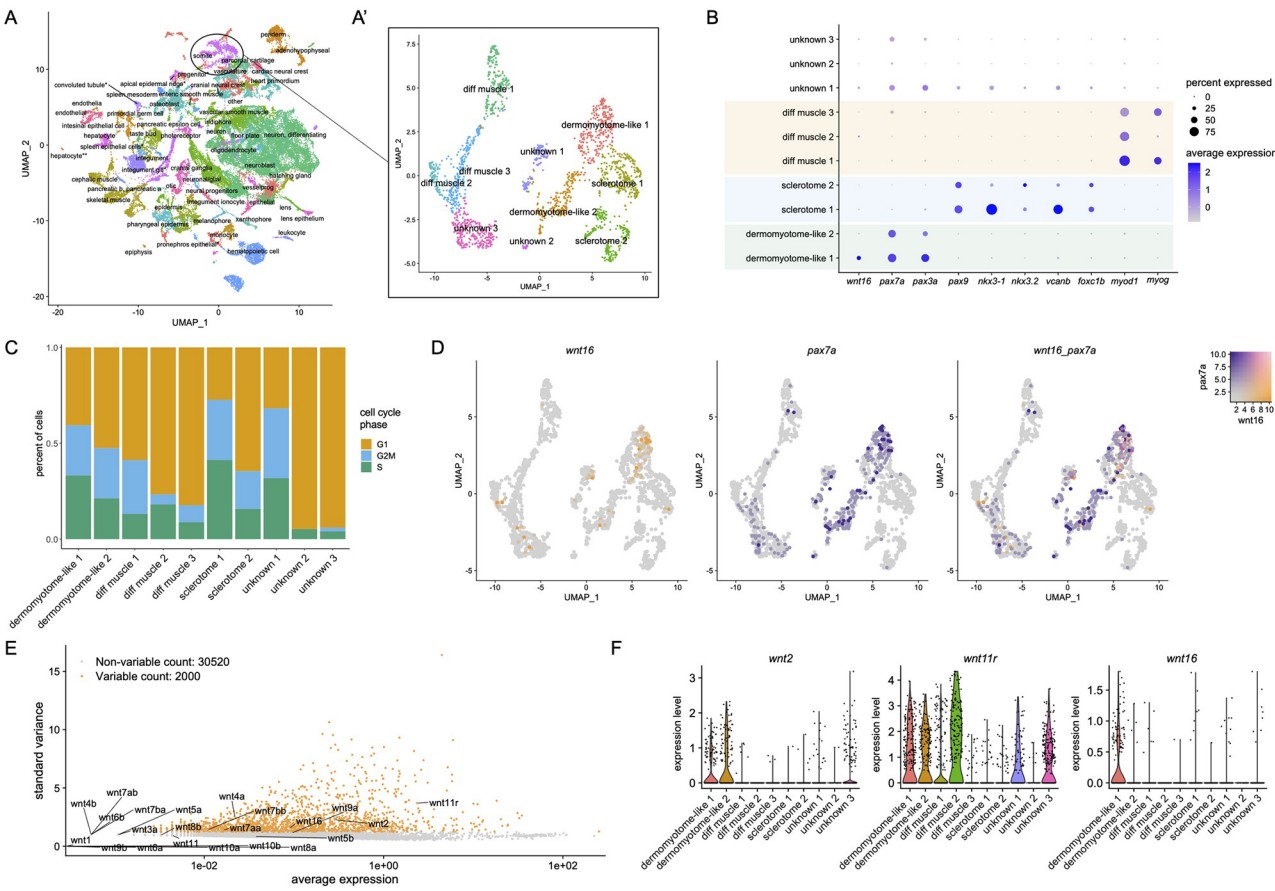

**Fig 1. Single-cell transcriptional profiling of somitic *wnt16*+ cells.** (A) UMAP plot visualizing cell clusters from the sci-RNA-seq analysis of zebrafish embryonic development, as performed by Farnsworth et al., Dev Biol 2020 459:100–108. Circle shows somite cluster used for analysis in this study. (A') Subclustering of the somite cluster. Of the 10 distinct clusters, 7 could be identified by molecular markers, including two dermomyotome-like populations. (B) Dotplot visualizing expression of genes demarcating the 12 cell clusters. Circle size represents the percentage of cells expressing the gene, and color indicates average gene expression. (C) Percentage of cells in each cell cycle phase. (D) Blend plots for *wnt16* and *pax7a*. (E) Variable features plot for all zebrafish *wnt* family members. Note that *wnt11* (*wnt11f1*) and *wnt11f2* are annotated in the atlas as *wnt11r* and *wnt11*, respectively. (F) Violin plots for *wnt2*, *wnt11* (annotated in the atlas as *wnt11r*), and *wnt16*.

We further characterized the *wnt16*+ cells within the dermomyotome-like 1 cluster. Cell cycle analysis revealed that the dermomyotome-like 1 cluster had a higher percentage of cells in the S or G2M phases compared to the dermomyotome-like 2 cluster (Fig 1C). A substantial number of *wnt16*+ cells were negative for *pax7a* (Fig 1D), suggesting that *wnt16*+ and *pax7a* + cells are only partially overlapping populations. Finally, we found that, along with *wnt2* and *wnt11* (also known as *wnt11f1*, annotated within the atlas as *wnt11r*) [29], *wnt16* was among the most highly and variably expressed *wnt* family members within the somite cluster (Fig 1E). Further analysis revealed that *wnt2* and *wnt11* (*wnt11r*) were differentially expressed in multiple clusters, in contrast to *wnt16*, whose expression was primarily confined to the dermomyotome-like 1 cluster (Fig 1F). Thus, *wnt16* specifically demarcates a subset of dermomyotome-like cells within the external cell layer.

At the time of our analysis of the scRNA-seq atlas, notochord cells were not annotated. However, in cells annotated as "other", we detected a high correlation between expression of *wnt16* and the early notochord marker *ntla/tbxta* (S1 Fig), further supporting that, in addition to dermomyotome, *wnt16* is expressed within the developing notochord.

## *wnt16* is expressed in dermomyotome and developing notochord

To place our scRNA-seq findings into a spatial context, we performed RNA *in situ* hybridization (ISH). In transverse sections of 22 hpf (hours post fertilization) embryos, *wnt16+* cells were visible within the lateral portion of the trunk, in a region likely containing the external cell layer and/or developing slow muscle (Fig 2A and 2B). Staining for *wnt16* in the lateral portion included expression within the presumptive external cell layer, which was visible as a single layer of cells superficial to the myotome (Fig 2C). Staining for *pax7a* was detectable in cells alongside *wnt16+* cells within the lateral portion of the trunk. Moreover, staining for *pax7a*, but not *wnt16*, was observed dorsal to the neural tube in the most anterior sections (Fig 2D). *wnt16+* and *pax7a+* cells were also sporadically found deep within the myotome. While instances of *wnt16* and *pax7a* co-localization existed, we frequently observed instances of

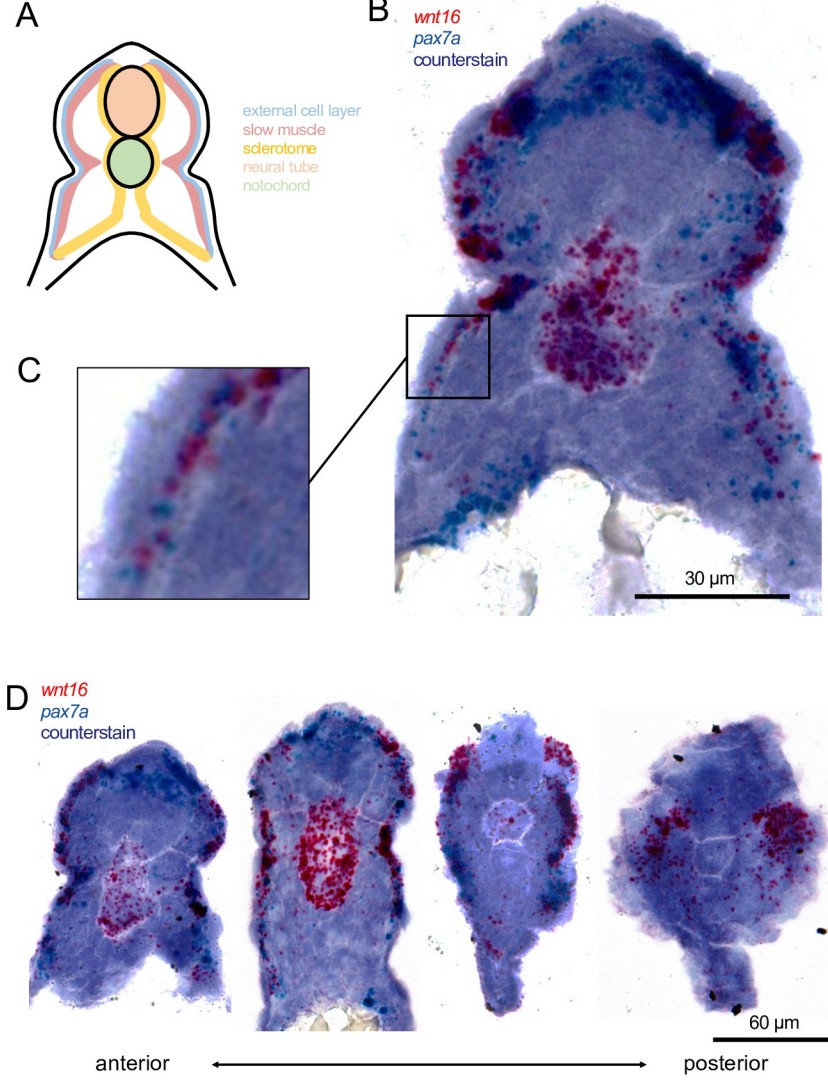

**Fig 2. *wnt16* is expressed in dermomyotome-like cells at 22 hpf.** (A) A schematic of a transverse section through the zebrafish trunk with compartments labeled and color coded. (B-C) Chromogenic in situ hybridization of an anterior (note the yolk in the ventral space) transverse section through the zebrafish trunk shows cells expressing *wnt16* and *pax7a* are located in the external cell layer (magnified in C). (D) Transverse sections, beginning in the anterior trunk (left), and moving posteriorly (right), representing less mature somites, show variation in *wnt16+* labeling.

independent *wnt16+* and *pax7a+* expression domains. Thus, *wnt16+* cells are expressed similarly to, but not totally overlapping with, *pax7a+* cells.

Strong staining for *wnt16* in 22 hpf embryos was also detected in the developing notochord. Staining was strongest in the anterior sections and weakest or absent in the posterior sections, suggesting transient expression in the anterior-to-posterior direction during notochord development as has been seen for *wnt11* [30]. Notochord expression has not been previously reported in prior zebrafish studies of *wnt16* in hematopoietic stem cell and spine development [13, 21].

## Isolation of *wnt16*$^{-/-}$ mutants

Previous studies have reported disparate skeletal phenotypes in zebrafish homozygous for presumptive *wnt16* null alleles; for example, Qu et al. reported loss of caudal fin rays, whereas caudal fin rays in mutants described by McGowan et al. are intact [13, 14]. To help resolve these discrepancies, we generated multiple *wnt16* loss-of-function alleles using CRISPR/Cas9-based gene editing (S2A Fig). We targeted exon 3 or exon 4 of the *wnt16* gene for the generation of indels and isolated three zebrafish mutant lines, *wnt16*$^{w1001}$, *wnt16*$^{w1008}$, and *wnt16*$^{w1009}$. *wnt16*$^{w1001}$ (c.518_521delinsGTCATTTATTTAAA) leads to an induced frameshift and premature stop codon in exon 3. *wnt16*$^{w1008}$ (c.639_654delCTGTCATGGCGTATCG) and *wnt16*$^{w1009}$ (c.628_642delinsCCGCTGTT) each lead to an induced frameshift and premature stop codon in terminal exon 4. All three alleles are predicted to result in early truncation of the Wnt16 protein and subsequent loss of function. Specifically, for all three alleles, the predicted early truncation results in loss of a highly conserved serine (S218) at the tip of the "thumb" in the N-terminal domain (S2B Fig) that is a site of Wnt acylation necessary for secretion and activity [31], as well as loss of the C-terminal "finger" domain that, together with the "thumb", act as the two major binding sites that allow grasping to the Frizzled cysteine-rich domain (CRD) [32].

We performed RT-PCR spanning the *wnt16* locus to assess the effects of *wnt16*$^{w1001}$ on *wnt16* transcript levels and splicing (S2C Fig). We did not observe evidence of the generation of novel splice variants by CRISPR-induced indels [33]. While *wnt16*$^{w1001}$ is predicted to activate nonsense-mediated decay (NMD), we observed no obvious reduction in *wnt16* transcript levels. *wnt16*$^{w1008}$ and *wnt16*$^{w1009}$ lead to a predicted premature stop codon in the terminal exon and thus are not predicted to activate NMD or associated genetic compensation [34]. Heterozygous incrosses produced larval progeny at roughly Mendelian ratios. Homozygous mutants for all three mutant alleles exhibited similar reductions in standard length and morphological abnormalities during vertebral development (S2D and S2E Fig). The close correspondence in mutant phenotypes as well as the predicted loss of key domains and residues critical for Wnt16 activity and secretion suggest that all three alleles are likely functioning as null alleles. We refer to all three mutants as *wnt16*$^{-/-}$. All three mutants are used throughout the study.

## *wnt16* is necessary for notochord and myotome morphogenesis

We assessed the necessity of *wnt16* for notochord and muscle development. RNA ISH in *wnt16*$^{-/-}$ mutants at 1 dpf (days post fertilization) revealed no obvious differences in expression of markers of muscle (*pax7a*, *myog*) and notochord (*ntla/tbxta*) differentiation (S3 Fig). At 3 dpf, *wnt16*$^{-/-}$ mutants exhibited a reduction in standard length (Fig 3A and 3B). We did not observe notochord lesions or obvious impairment of formation of notochord vacuoles (Fig 3A inset). No significant reductions in body length were observed at 1 or 2 dpf (Fig 3B), suggesting that *wnt16* contributes to axial elongation of segmented tissue in post-tailbud stages. In

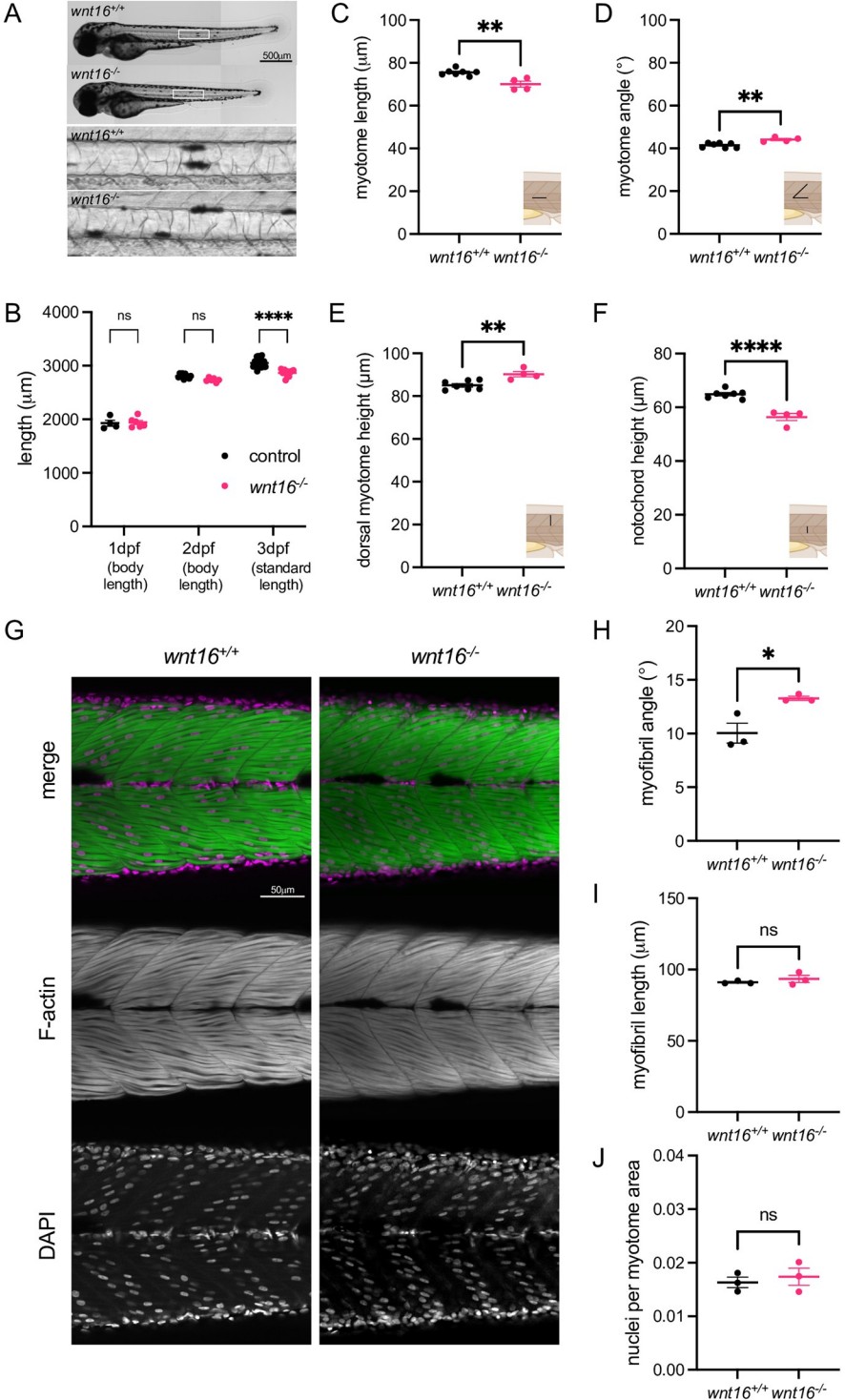

**Fig 3. *wnt16* is required for notochord and myotome morphogenesis.** (A) Brightfield images of *wnt16*[-/-] mutants and clutchmate controls. Insets show closeup of notochord. (B) *wnt16*[-/-] mutants exhibit significantly reduced standard length at 3 dpf. (C-F) Myotome morphology (C-E) and notochord height (F) are altered in *wnt16*[-/-] mutants. Insets provide schematic of measurement. (G) Lateral views of phalloidin-stained 3dpf animals obtained using confocal microscopy. (H-I) Quantitative image analysis reveals that myofibril angle (H) but not myofibril length (I) or nuclei per myotome area (J) are altered in *wnt16*[-/-] mutants. P-values were determined using either two-way ANOVA with Fisher's LSD post hoc test (B) or an unpaired t-test (C-F, H-J). *p<0.05, **p<0.01, ****p<0.0001, ns: not significant.

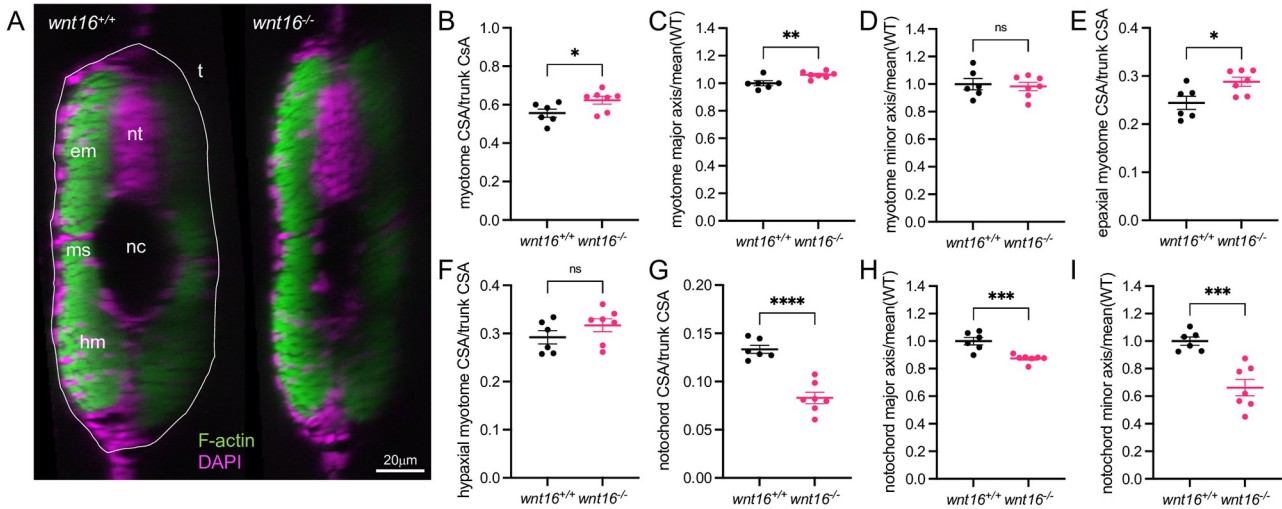

**Fig 4. *wnt16* suppresses myotome expansion and promotes notochord radial expansion.** (A) Transverse views of phalloidin-stained 3dpf animals. em: epaxial myotome, hm: hypaxial myotome, ms: myosepta, nc: notochord, nt: neural tube, t: trunk. (B-I) Quantification of myotome (B-F) and notochord (G-I) morphology. P-values were determined using an unpaired t-test. *p<0.05, **p<0.01, ***p<0.001, ****p<0.0001, ns: not significant.

lateral views, 3 dpf *wnt16*[-/-] mutants exhibited significantly reduced myotome length (Fig 3C) and altered myotome boundary angle (Fig 3D). Dorsal myotome height was also significantly increased in 3 dpf *wnt16*[-/-] mutants (Fig 3E), whereas notochord height was significantly reduced (Fig 3F). Similar phenotypes were observed in zygotic and maternal zygotic *wnt16*[-/-] mutants (S4 Fig), suggesting that maternal stores of wild-type transcript/protein were not masking potentially more severe phenotypes. 3D imaging of phalloidin-stained animals (Fig 3G) revealed that myofibril angle was altered in *wnt16*[-/-] mutants (Fig 3H). No significant differences in myofibril length or number of nuclei per myotome were observed (Fig 3I and 3J).

Next, we examined whether *wnt16* contributes to specific myotome compartments or developmental axes. A recent study in medaka demonstrated that Wnt11 acts on dermomyotome cells to guide epaxial myotome morphogenesis [35]. Analysis of transverse sections of phalloidin-stained animals revealed that *wnt16*[-/-] mutants exhibited an increase in normalized myotome cross-sectional area (CSA) (Fig 4A and 4B). Further, *wnt16*[-/-] mutants exhibited an increase in normalized myotome major axis (approximately along the dorsal-ventral axis) whereas no significant difference was observed in normalized minor axis (Fig 4C and 4D). Normalized epaxial myotome CSA was significantly increased in *wnt16*[-/-] mutants (Fig 4E). No significant difference was observed for normalized hypaxial myotome CSA in *wnt16*[-/-] mutants, however, there was a trend toward this being increased (Fig 4F). Moreover, normalized notochord CSA was decreased in *wnt16*[-/-] mutants (Fig 4G) with significant reductions observed in both normalized major and minor axes (Fig 4H and 4I). These data suggest that, in the notochord, *wnt16* promotes notochord length and radial expansion; in the myotome, *wnt16* suppresses dorsal-ventral elongation and myotome expansion with apparently greater effects in epaxial myotome.

## *wnt16* is expressed in the ventral midline of the notochord sheath

We next asked how *wnt16* contributes to spine formation. While *wnt16* expression was relatively absent in osteoblastic precursors within sclerotome, the expression of *wnt16* in developing notochord in embryos brought forth the question of whether *wnt16* is expressed in the notochord in late larvae when osteoblastic cells condense around the mineralizing notochord.

We performed RNA ISH to examine *wnt16* expression in 12 dpf larvae (WT: 5.4mm SL), a stage when notochord sheath domains are mineralizing, and condensation of osteoblastic cells from sclerotome around the notochord has presumably initiated. Co-staining for *wnt16* and *pax7a* in transverse sections (Fig 5A) revealed staining for *wnt16* in cells distributed along the lateral portion of the myotome (Fig 5A inset, long arrow), within or adjacent to presumptive myosepta (Fig 5A inset, short arrow), and sporadically within the myotome (Fig 5A inset, arrowhead), which also contained cells staining for *pax7a*. The strongest staining for *wnt16* was detected in the notochord (Fig 5A, star), and was mostly negative for *pax7a*. In contrast to embryos where *wnt16* was uniformly detected throughout the cross-section of the developing notochord, in larvae, *wnt16* staining was spatially restricted to the ventral midline, within or adjacent to the notochord sheath.

To help distinguish whether *wnt16*+ cells were notochord sheath cells or osteoblastic cells condensing around the notochord, we performed co-staining for *cdh11*, a marker of sclerotome in mouse embryos [36] and mesenchymal-like osteoblastic cells in zebrafish [37] (Fig 5B). Staining for *cdh11* was apparent in several locations associated with sclerotome domains, including in cells surrounding the notochord (Fig 5B, inset), the dorsal myotome (Fig 5B, long arrow), presumptive myosepta (Fig 5B, short arrow), and the medial portion of the ventral myotome (Fig 5B, arrowhead) [28]. Notably, staining for *wnt16* was localized within a single layer of cells surrounded by cells staining positively for *cdh11* Fig 5B, inset). Wopat et al. previously showed that notochord sheath cells can be distinguished into three domains: a mineralizing domain (*entpd5a*+), a non-mineralizing domain that eventually demarcates the intervertebral disc (*col9a2*+), and a transitional domain (*entpd5a*+/*col9a2*+). Analysis of previous RNA-seq data generated from FACS-sorted *entdp5a*+ and *col9a2*+ notochord populations

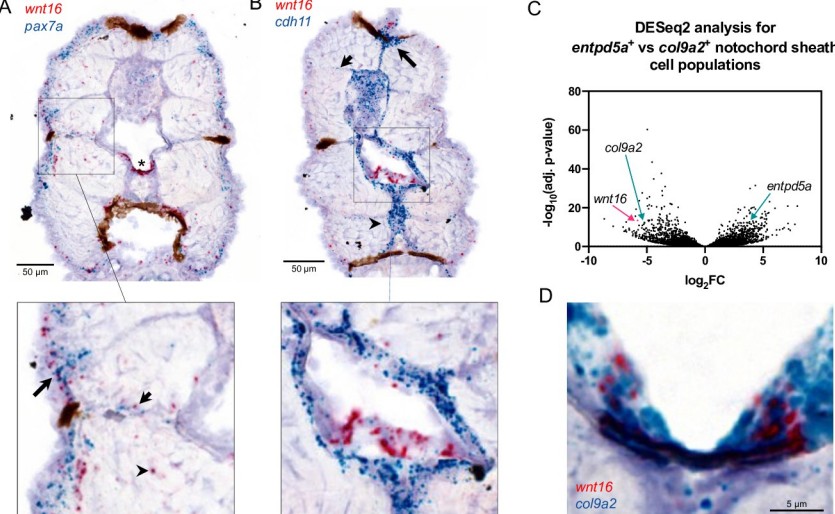

**Fig 5. *wnt16* is expressed in the ventral midline of the notochord sheath.** (A, B) Chromogenic *in situ* hybridization was performed in transverse sections through the zebrafish trunk at 12 dpf. (A) Co-staining for *wnt16* and *pax7*. Staining for *wnt16* was detected in notochord (star) and was mostly negative for *pax7a*. Staining for *wnt16* was also detected in cells in the lateral portion of the myotome (inset, long arrow), within or adjacent to presumptive myosepta (inset, short arrow), and sporadically within the myotome (inset, arrowhead). (B) Co-staining for *wnt16* and *cdh11*. Staining for *cdh11* was apparent in cells surrounding the notochord, the dorsal myotome (long arrow), presumptive myosepta (short arrow), and the medial portion of the ventral myotome (arrowhead). With regard to notochord, staining for *wnt16* was localized within a single layer of cells surrounded by cells staining positively for *cdh11* (inset). (C) Analysis of RNA-seq data from [18] shows that *wnt16* is differentially expressed in *col9a2*+ relative to *entpd5a* + sheath cells. (D) Co-staining for *wnt16* and *col9a2* shows co-localization at the ventral midline of the notochord sheath.

[18] revealed *wnt16* was strongly expressed in the *col9a2+* but not the *entpd5a+* population (Fig 5C). Consistent with RNA-seq analyses, RNA ISH revealed co-staining for *wnt16* in *col9a2+* cells at the ventral midline of the notochord sheath (Fig 5D). Taken together, these studies indicate that *wnt16* is expressed at the ventral midline of the notochord sheath during spine development.

## *wnt16* is necessary for notochord sheath mineralization

To examine the relationship between *wnt16* expression in the ventral midline of the notochord sheath and vertebral development, we performed calcein staining in *wnt16*−/− larvae. In wild-type animals (WT: 5.7mm SL), calcein staining was detectable in most vertebrae, appearing as rectangularly shaped domains in the lateral view (6A, top). Moreover, we often observed a single thin line of staining at the ventral notochord at presumptive sites of mineralizing domains (Fig 6A, top inset). We interpreted these ventral lines of staining to indicate that zebrafish chordacentrum mineralization initiates at the ventral midline of the notochordal sheath and

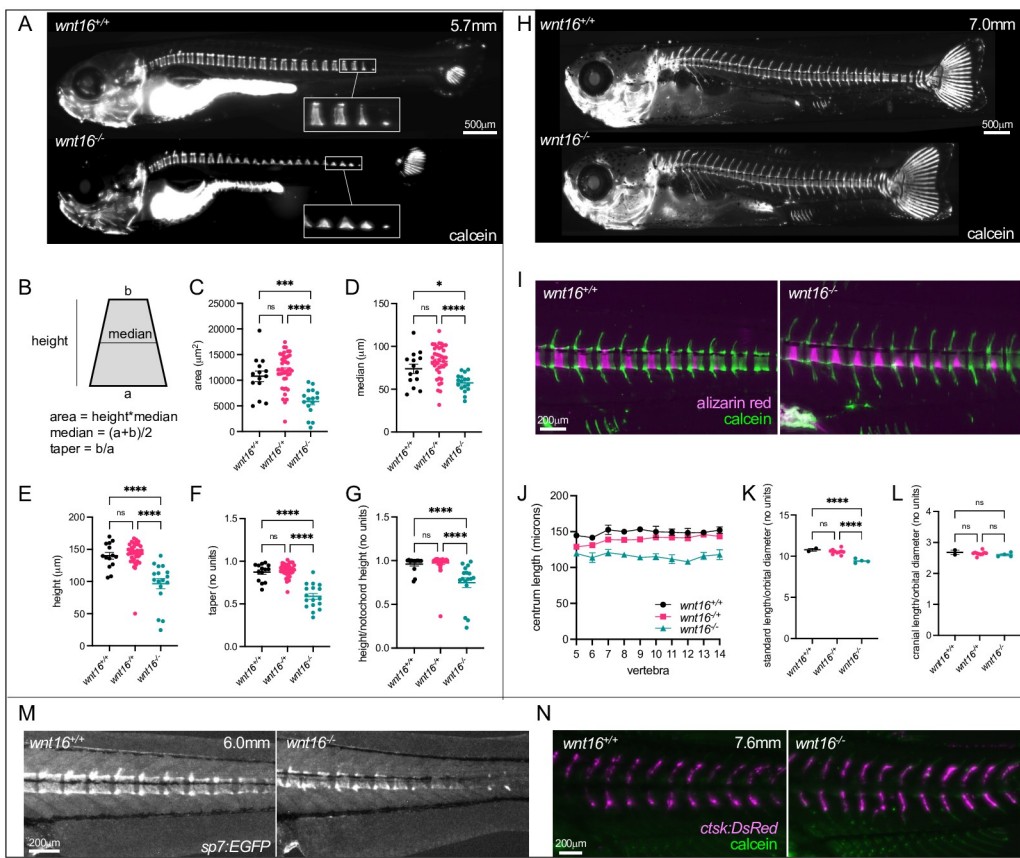

**Fig 6. *wnt16* is necessary for notochord sheath mineralization.** (A) Calcein staining in *wnt16*−/− larvae (WT: 5.7mm SL). In *wnt16*−/− mutants, mineralized domains are irregular in shape, incompletely proceeding to the dorsal surface from the ventral surface of the notochord. (B-G) Quantification of mineralizing domain morphology. In (C-G), each point represents a single fish, with values averaged from vertebrae 1–16. (H) Calcein staining in *wnt16*−/− larvae (WT: 7.0mm SL). (I) Dual fluorochrome staining shows trapezoid-like mineralized domains in *wnt16*−/− mutants eventually became rectangle-like. (J-L) Quantification of centrum length (J), standard length (K), and cranial length (L). (M) Visualization of osteoblastic cells using *Tg[sp7:EGFP]*. EGFP domains in mutants appear trapezoid-like with reduced expression on the dorsal surface, resembling calcein domains in (A). (N) Visualization of osteoclastic cells using *Tg[ctsk:DsRed]*. Similar DsRed domains are observed in wildtype and *wnt16*−/− mutants. P-values were determined using one-way ANOVA with Fisher's LSD post hoc test. *p<0.05, **p<0.01, ***p<0.001, ****p<0.0001, ns: not significant.

proceeds toward the dorsal side of the notochord before forming a complete mineralized ring, similar to salmon [38]. In *wnt16*$^{-/-}$ mutants, while calcein staining was detectable in most vertebrae, domains appeared irregular in shape, incompletely proceeding to the dorsal surface from the ventral surface of the notochord (Fig 6A, bottom inset). Quantification of mineralizing domains revealed that vertebrae in *wnt16*$^{-/-}$ mutants were smaller in area due to decreased height and width, and had an altered shape with a more trapezoid-like (as opposed to rectangle-like) appearance (Fig 6B–6F). Moreover, both the notochord height and the ratio of the mineralizing domain height to notochord height was reduced in *wnt16*$^{-/-}$ mutants (Fig 6G). Thus, we conclude that *wnt16* is dispensable for initial chordacentrum mineralization at the ventral midline, but necessary for patterning chordacentrum width, and for mineralization to proceed in the dorsal direction.

In more mature animals (WT: 7.0mm SL), calcein staining was detectable in all vertebrae (Fig 6H). Dual fluorochrome staining revealed that trapezoid-like mineralized domains in *wnt16*$^{-/-}$ mutants eventually became rectangle-like, demonstrating the potential for developing chordacentra in mutants to recover a ring-like shape (Fig 6I). Centrum length was reduced in *wnt16*$^{-/-}$ mutants (Fig 6J). Segmentation defects were not evident; specifically, we did not observe loss of vertebral size uniformity, or obvious differences in prevalence of vertebral fusions in *wnt16*$^{-/-}$ mutants. While standard length was reduced in *wnt16*$^{-/-}$ mutants (Fig 6K), cranial length was unaffected (Fig 6L), indicating that shortening was specific to the post-cranial skeleton. Prior studies have shown that notochord sheath mineralization is necessary for recruitment of osteoblastic cells, and provides a template for the formation of mature centra [18]. We used the transgenic line *Tg[sp7:EGFP]* to visualize osteoblastic cells [39]. We found that EGFP in *wnt16*$^{-/-}$ mutants was disrupted in developing centra; in contrast to the rectangle-like (ring-like in 3D) domains in wildtype fish (WT: 6.0mm SL), EGFP domains in mutants appeared trapezoid-like, with reduced expression on the dorsal surface (Fig 6M), and resembled calcein domains (Fig 6A). Thus, *wnt16* is necessary for osteoblastic recruitment to developing vertebrae, potentially through its effect on notochord sheath patterning and mineralization.

We also examined whether *wnt16* is necessary for recruitment of osteoclastic cells. Previous studies have shown that *Wnt16* KO mice exhibit increased bone resorption, owing to a function of WNT16 in suppressing osteoclastogenesis [12]. We used the transgenic line *Tg[ctsk: DsRed]* to visualize osteoclastic cells [40]. Similar DsRed domains were observed in wildtype (WT: 7.6mm SL) and *wnt16*$^{-/-}$ mutants (Fig 6N); for both groups, expression was largely absent from centra and localized to the neural and haemal arches. Thus, *wnt16* is not necessary for osteoclastic recruitment to developing vertebrae. Taken together, our findings indicate that, in addition to being necessary for notochord and myotome morphology in embryos, *wnt16* is required for spine morphogenesis in larvae.

## *wnt16* is necessary for adult bone mass and morphology

Variants at the *CPED1-WNT16* locus are associated with BMD in both pediatric and adult populations, indicating they have an impact throughout the lifespan. Thus, we asked whether *wnt16* is necessary for spine and muscle mass or morphology in adult animals. We performed microCT scanning in *wnt16* mutants and wildtype clutchmates at 90 dpf. Scans were used for spinal phenomic profiling using FishCuT software [24]. For this, we computed ten combinatorial quantities (the nine possible combinations of 3 vertebral elements (centrum, Cent; neural arch, Neur; and haemal arch, Haem) x 3 characteristics (tissue mineral density, TMD; thickness, Th; and volume, Vol) plus centrum length (Cent.Le) in the 20 anterior-most pre-caudal and caudal vertebrae. For each combination of outcome/element, we computed a standard

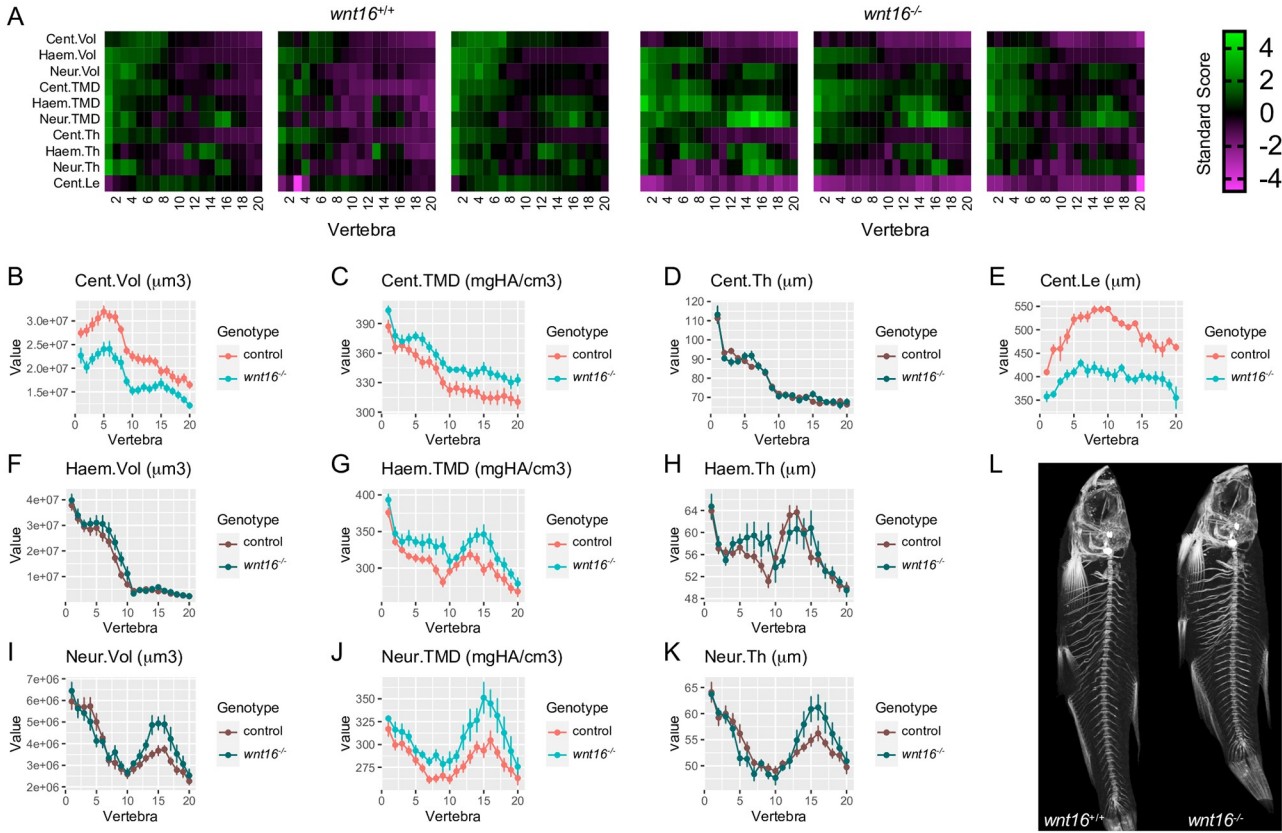

**Fig 7. Adult *wnt16^-/-* mutants exhibit reduced centrum length and increased mineralization.** (A) Skeletal barcodes visually depict individual phenomes for *wnt16^+/+* and *wnt16^-/-* fish (3 fish/group shown). (B–K) Phenotypic features, indicated by the graph title (with units for y axis), are plotted as a function of vertebra along the spine. Control indicates *wnt16^+/+ or wnt16^-/+* clutchmates. Measures with p<0.05 in the global test are in a lighter coloring scheme, n = 6-10/group. Cent, centrum; Haem, haemal arch; Neur, neural arch; Vol, volume; TMD, tissue mineral density; Th, thickness; Le, length. (L) Maximum intensity projections of microCT scans.

score and arranged these data into matrix constructs referred to as "skeletal barcodes" (Fig 7A). The skeletal barcodes for *wnt16^-/-* indicated a reduction in centrum volume and length (note the dark/purple color on top and bottom rows in *wnt16^-/-* barcodes) and an increase in centrum, haemal arch, and neural arch TMD.

We plotted these quantities as a function of vertebra number along the axial skeleton and calculated p-values using the global test [24]. We found no significant differences between wildtype fish and heterozygous clutchmates (S5 Fig), thus *wnt16^+/+ and wnt16^-/+* fish were pooled into a single control group. Analysis revealed that *wnt16^-/-* mutants exhibited significant differences in centrum volume and centrum length, as well as centrum, haemal arch, and neural arch TMD (Fig 7B, 7C, 7E, 7G and 7J), compared to the control group. WNT16 has been previously shown to suppress osteoclastogenesis in mice [12]. We observed no differences in thickness of the neural arch, a compartment enriched with osteoclasts in zebrafish and medaka and whose morphological changes indicate alterations in osteoclast activity [40, 41]. However, we did observe changes in neural arch angle (S9D Fig), a measure previously shown to be sensitive to changes in osteoclast activity [40, 42]. We also found *wnt16^-/-* fish exhibited reduced standard length compared to controls (S6 Fig). An allometric scaling analysis was also performed as described in Hur et al. [24] to explore the potential that phenotypic changes may be attributable to differences in developmental progress. The analysis indicated

significant changes in measures even when they were normalized by standard length, indicating that mutant phenotypes in $wnt16^{-/-}$ fish are not solely attributable to developmental delay (S7 Fig). No obvious differences in number of vertebrae, rib fracture calluses, neural arch non-unions, centrum compressions, or centrum fusions were observed in $wnt16^{-/-}$ mutants. Taken together, these data indicate that $wnt16$ impacts adult vertebral bone mass and morphology.

## *wnt16* is necessary for adult lean mass and morphology

We next assessed the necessity of $wnt16$ for adult muscle morphology. We implemented a procedure for fully automatic calculation of pixel intensity thresholds and independent segmentation of lean and bone tissue from microCT scans, similar to an approach used previously [43]. MicroCT scans of adult zebrafish occasionally revealed two distinct compartments of soft tissue with high and low attenuation coefficients. In small rodents, such regions have been attributed to skeletal muscle and adipose tissue, respectively [44]. Comparison of areas of tissue derived via automatic and manual segmentation revealed a high correlation for lean and bone tissue (S8 Fig). Lower correlation was obtained for presumptive adipose tissue (S8 Fig). Because of the reduced accuracy in discriminating between skeletal muscle and presumptive adipose tissue, and the fact that presumptive adipose tissue was not uniformly visible in wild-type and mutant fish, presumptive adipose tissue was included in calculations of lean tissue volume. When visible, we estimate that presumptive adipose tissue was a minor component (~10%) of lean tissue volumes, and comparable to values computed elsewhere [45]. Processed images were used to visualize and quantify lean mass independently of bony tissue (Fig 8A).

$wnt16^{-/-}$ mutants showed noticeable changes in body shape at 90 dpf. Accordingly, we calculated the fineness ratio (the ratio of standard length to dorsoventral height), a measure of body shape that correlates with swimming speed in coral reef fishes [46], and found this was reduced in $wnt16^{-/-}$ mutants (Fig 8B). Analysis of lean tissue in adult animals revealed $wnt16^{-/-}$ mutants exhibited an altered distribution of lean tissue along the anterior-posterior axis (Fig 8C). Specifically, there was a trend toward increased lean cross-sectional area in the anterior trunk, and decreased cross-sectional area in the posterior trunk in the $wnt16^{-/-}$ mutants. This was evident by comparing the topology of the curves and the presence of a local spike corresponding to the boundary between the anterior and posterior swim bladders (arrowhead, Fig 8C). While the shapes of the curves were similar anterior to this spike, posterior to this point, there was a visible decrease in cross-sectional area and shortening of the trunk. Trunk lean volume and anterior trunk lean volume (i.e. the lean volume in the trunk anterior to the anterior/posterior swim bladder boundary) in $wnt16^{-/-}$ mutants was similar to controls (Fig 8D). In contrast, posterior trunk lean volume (i.e. the lean volume in the trunk posterior to the anterior/posterior swim bladder boundary) was significantly decreased in $wnt16^{-/-}$ mutants (Fig 8D). Moreover, whereas the anterior swim bladder was similarly sized between $wnt16^{+/+}$ and $wnt16^{-/-}$ fish, the posterior swim bladder was shorter in $wnt16^{-/-}$ mutants (Fig 8E).

Next, we asked if the shape and size of myomere segments might be affected in $wnt16^{-/-}$ mutants. Due to the challenges associated with analyzing complex three-dimensional myomere morphology, we first assessed measures from the axial skeleton which are correlated with measures of myomere shape and size [47]. Specifically, centrum length, neural arch length and neural arch angle were used to provide correlates for myomere length, myomere height and myomere angle, respectively (S9A–S9D Fig). The $wnt16^{-/-}$ mutants showed a reduction in centrum length and an increase in neural arch angle, but no change in neural arch length across the spine, suggesting a shift toward a more narrow, rectangular myomere shape in the $wnt16^{-/-}$ fish. To validate the myomere phenotypes inferred by skeletal correlates, fish were subjected to contrast-enhanced high resolution microCT [48]. As predicted, increased myomere angle and reduced

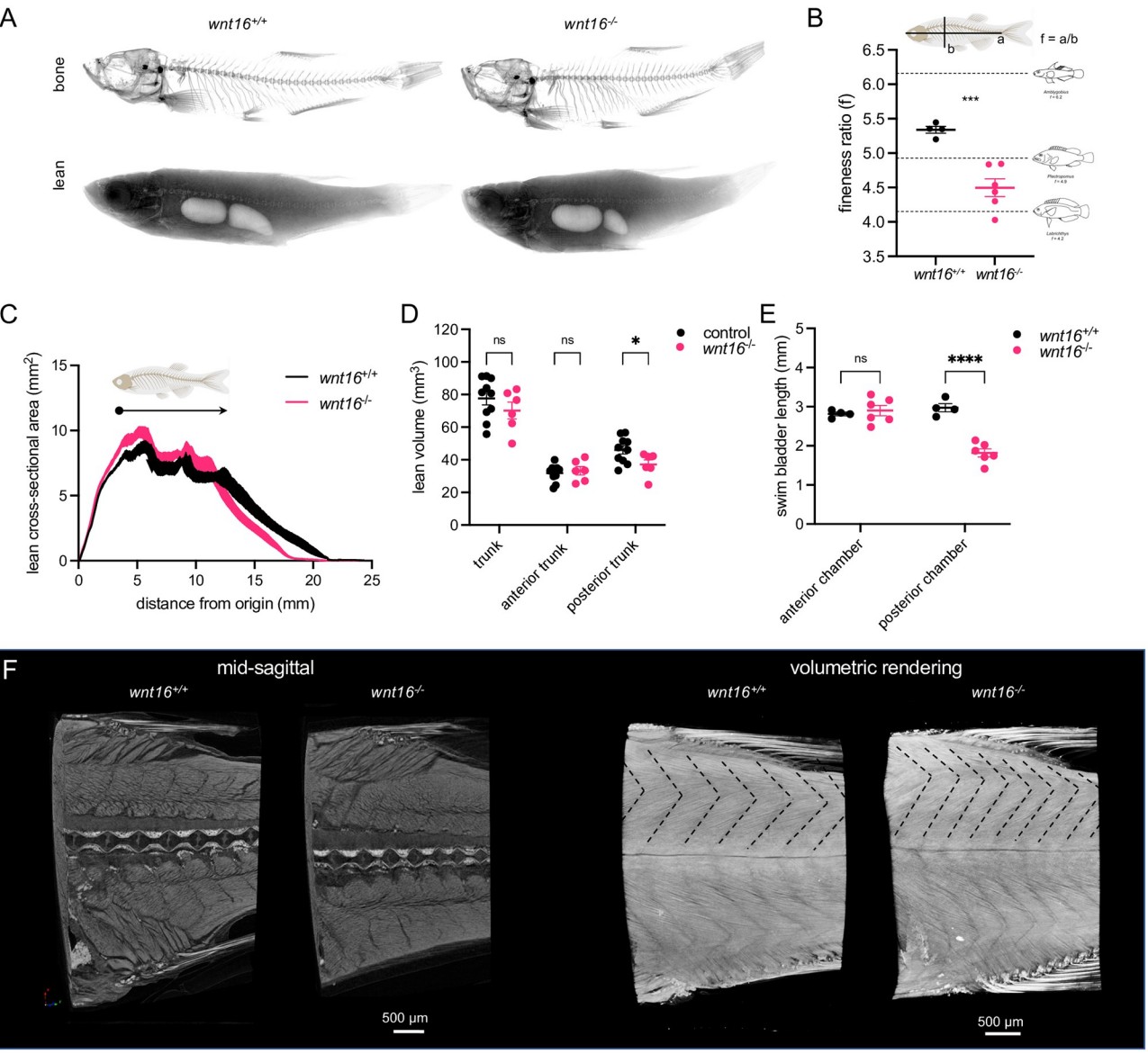

**Fig 8. Adult *wnt16*⁻/⁻ mutants exhibit altered lean mass and morphology.** (A) Segmentation of microCT images for bone (top) and lean (bottom) tissue. Shown are average intensity projections. (B) *wnt16*⁻/⁻ mutants exhibit reduced fineness ratio. Fish images adapted from [46]. (C) Lean cross-sectional area as a function of distance along the anteroposterior axis. Thickness of the line indicates standard error of the mean (n = 4-6/group). Arrowhead indicates approximate boundary between the anterior and posterior swim bladders. (D) Lean volume in the trunk, anterior trunk, and posterior trunk. *wnt16*⁻/⁻ mutants exhibit reduced lean volume in the posterior trunk compared to controls (*wnt16*⁺/⁺ and *wnt16*⁺/⁻) (E) *wnt16*⁻/⁻ mutants exhibit reduced swim bladder length in the posterior, but not anterior, chamber compared to wildtype. (F) High-resolution contrast-enhanced microCT reveals altered myomere width and angle in *wnt16*⁻/⁻ mutants as indicated by dotted lines in the top part of the myotome compartment in volumetric renderings. P-values were determined using an unpaired t-test (B) or two-way repeated measures ANOVA (D and E) with Fisher's LSD post hoc test. *p<0.05, ***p<0.001, ****p<0.0001. Fig 8 adapts portions of Fig 1 from the following paper: Walker, J.A., Alfaro, M.E., Noble, M.M., and Fulton, C.J. (2013). Body fineness ratio as a predictor of maximum prolonged-swimming speed in coral reef fish. PLoS One 8, e75422. The paper of Walker et al., which was published in PLoS One, applies the Creative Commons Attribution 4.0 International (CC BY) license (https://journals.plos.org/plosone/s/licenses-and-copyright).

myomere width in *wnt16*⁻/⁻ mutants were indicated in volumetric renderings (Fig 8F). Taken together, we have shown that *wnt16* is necessary for adult lean tissue mass and morphology.

Finally, we explored whether loss of *wnt16* results in muscle pathology. H&E-stained sections in juvenile (30 dpf) *wnt16*⁻/⁻ mutants revealed apparently normal myomere boundaries

and no obvious differences in muscle fiber morphology compared to wildtype clutchmates (S10A Fig). In transverse sections in adult (>8 months post fertilization) animals, muscle fibers in *wnt16⁻/⁻* mutants appeared to be relatively normal in shape and size with peripheral nuclei as expected, and no obvious fiber degeneration or inflammatory infiltrates were detected (S10B Fig). Moreover, *wnt16⁻/⁻* mutants did not exhibit obvious swimming abnormalities and both mutants and wildtype clutchmates exhibited full-length body flexions when the startle-induced C-start response was invoked (S10C Fig). Thus, *wnt16⁻/⁻* mutants are absent of obvious muscle pathology, contractile dysfunction, or fiber degeneration.

## *wnt16* is a gene of major effect at the *CPED1-WNT16* locus

Our interest in *wnt16* as a candidate causal gene underlying pleiotropy at the *CPED1-WNT16* locus was based on data that supported *WNT16* as a causal gene underlying locus associations with osteoporosis-related traits [9, 10]. As described earlier, this chromosome region comprises two variants independently associated with BMD, one near *WNT16*, the other near *CPED1*. Because there is evidence these variants act on different regulatory elements and thus affect different genes [25], we asked if other genes at the *CPED1-WNT16* locus are necessary for lean mass or morphology.

We analyzed somatic mutants for genes at the *CPED1-WNT16* locus. In a forthcoming report, we describe a reverse genetic screen for causal genes underlying BMD-associated loci. As part of this screen, we generated somatic zebrafish mutants for five orthologous genes at the *CPED1-WNT16* locus: *tspan12*, *ing3*, *cped1* and *fam3c* in addition to *wnt16* (Fig 9A). Here, we asked whether these somatic mutants exhibited changes in lean tissue mass or morphology.

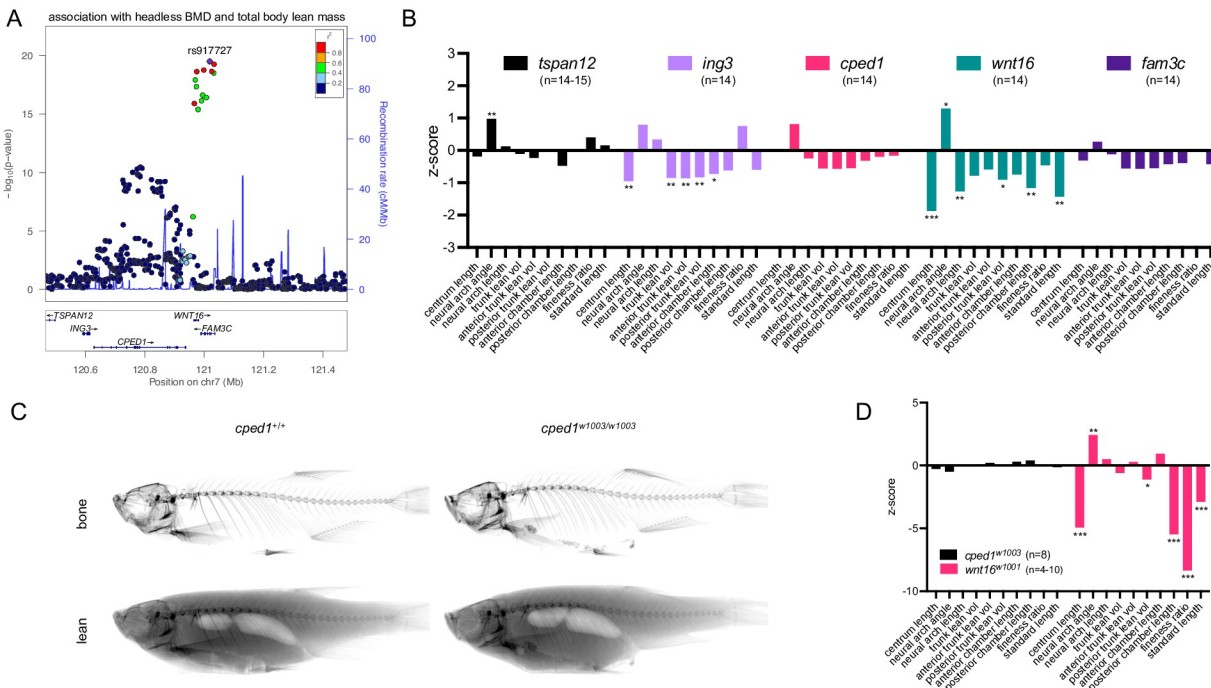

**Fig 9. *WNT16* is a gene of major effect on lean mass at the *CPED1-WNT16* locus.** (A) Schematic depicting variant associations and all genes with transcriptional start sites within ±500kb of the most significantly associated SNP at 7q31.31. (B) Z-scores for somatic mutants for *tspan12*, *ing3*, *cped1*, *wnt16*, and *fam3c*. (C) Segmentation of microCT images for *cped1*ʷ¹⁰⁰³ mutants for bone (top) and lean (bottom) tissue. (D) Z-scores for *cped1*ʷ¹⁰⁰³ and *wnt16*ʷ¹⁰⁰¹ mutants. P-values were determined using an unpaired t-test with the number of fish per group provided in the figure. *p<0.05, **p<0.01, ***p<0.001.

To assess this, mutants were microCT scanned at 90 dpf and scans were analyzed for measures of lean mass and morphology. Average mutation efficiencies ranged from approximately 40–80% (*tspan12*: 51.8%, *ing3*: 82.4, *cped1*: 39.7%, *wnt16*: 60.5%, *fam3c*: 50.0%). Similar to germline *wnt16*$^{-/-}$ mutants, somatic *wnt16* mutants showed significant differences in lean tissue correlates (reduced centrum length, increased neural arch angle, and reduced neural arch length), soft tissue morphology (posterior trunk lean volume, poster swim bladder chamber length), and body size (reduced standard length) (Fig 9B). These data support the potential for zebrafish somatic mutants to replicate phenotypic differences observed in germline mutants [26]. We also observed significant, though more muted, differences for somatic mutants for *ing3* (reduced centrum length, reduced trunk/posterior trunk/anterior trunk lean volume, reduced anterior chamber length) and *tspan12* (increased neural arch angle) (Fig 9B). Somatic mutants for *wnt16* exhibited the highest average z-score magnitude (*tspan12*: 0.27, *ing3*: 0.73, *cped1*: 0.35, *wnt16*: 1.1, *fam3c*: 0.36), the most number of significantly different measures (Fig 9B), and the highest average z-score magnitude normalized by average mutation efficiency (*tspan12*: 0.52, *ing3*: 0.89, *cped1*: 0.88, *wnt16*: 1.8, *fam3c*: 0.72), supporting the notion that wnt16 is a gene of major effect at the locus.

To further examine whether *wnt16* is a gene of major effect underlying pleiotropy at the *CPED1-WNT16* locus, we generated germline mutants for *cped1*, as *CPED1* is actively being investigated as a causal gene underlying effects at the *CPED1-WNT16* locus on osteoporosis-related traits [49]. We isolated *cped1*$^{w1003}$ (c.831-844delinsACT), which leads to an induced frameshift and premature stop codon in exon 6, which is predicted to result in early truncation of the Cped1 protein (Q277L;fsTer5) and loss of function. The mutation site maps to a site upstream of genomic sequences encoding for predicted Cadherin and PC-Esterase domains in mouse Cped1 (S11A Fig). While *cped1*$^{w1003}$ is predicted to activate NMD, RT-PCR spanning the *cped1* locus revealed no obvious reduction in *cped1* transcript levels (S11B Fig). No evidence of novel splice variants was observed (S11B Fig). MicroCT imaging in adult animals revealed apparently normal bone and lean tissue morphology in *cped1*$^{w1003}$ germline mutants (Fig 9C). Lean tissue phenotyping revealed no differences between germline *cped1*$^{w1003/w1003}$ mutants and wildtype controls, in contrast to multiple significant differences observed in *wnt16*$^{-/-}$ germline mutants (Fig 9D, S12 Fig). We observed no significant difference in phenotypes for both *cped1* somatic and *cped1*$^{w1003}$ mutants compared to their respective controls, further supporting the notion that *cped1* is dispensable for lean tissue mass and morphology. Taken together, these data support *WNT16* as a gene of major effect at the *CPED1-WNT16* locus.

## Discussion

This study identifies a dual requirement of *wnt16* for spine and muscle morphogenesis. *wnt16* appears to contribute to spine morphogenesis in at least two ways. First, *wnt16* is expressed in the developing notochord and promotes notochord elongation, which influences spine length later in development. Second, *wnt16* is expressed in the ventral midline of the notochord sheath, and promotes notochord sheath mineralization and subsequently, osteoblast recruitment. In regard to its function in muscle morphogenesis, Wnt16 is expressed in dermomyotome, where it appears to act as a morphogenetic signal on the adjacent developing myotome, as evidenced by the myotome elongation along the dorsal-ventral axis and expanded epaxial myotome in *wnt16*$^{-/-}$ mutants. This work expands our understanding of *wnt16* in musculoskeletal development and supports the potential for variants that act through *WNT16* to exert dual effects on bone and muscle via parallel morphogenetic processes.

Our results identifying *wnt16* expression in dermomyotome elaborate on previous work in zebrafish demonstrating expression of *wnt16* in the anterior somite [21], and are consistent

with prior studies of WNT function in embryonic myogenesis. These studies have found that secreted WNT glycoproteins from adjacent structures signal to the developing muscle tissue where they function as morphogenetic cues [50]. Several WNT family members have been identified to be critical for this process including WNT1, 3A and 4 which are secreted from the dorsal regions of the neural tube; WNT4, 6, and 7A which are secreted from the dorsal ecto-derm, and WNT11 which is secreted from the epaxial dermomyotome [50, 51]. While the der-momyotome is known to give rise to muscle and dermis, studies in medaka showed that Wnt11 regulates a non-myogenic, dorsal dermomyotome population that guides the dorsal expansion of the epaxial myotome [35]. In our studies, effects of loss of *wnt16* were stronger in expanding the epaxial myotome relative to the hypaxial myotome. This, in concert with our findings indicating that *wnt16* suppresses dorsal-ventral myotome elongation, suggests dermo-myotome-derived Wnt16 may potentially antagonize the dorsalizing influence of Wnt11 on epaxial myotome.

In regard to the expression of *wnt16* in notochord, several processes may contribute to notochord, myotome, and spine pathogenesis. We found that in embryos, notochord expres-sion of *wnt16* was reminiscent of that previously reported for *wnt11* (formerly known as *wnt11r*), which is expressed transiently in the notochord in an anterior-to-posterior direction during notochord development [30]. It is possible that notochord-derived Wnt16 and Wnt11 act in concert to influence notochord morphology. In zebrafish, notochord cells contribute to axial elongation by generating hydrostatic pressure within the notochord sheath, providing mechanical support for surrounding soft tissue prior to spine formation [52, 53]. Previous studies in zebrafish mutants have linked notochord lesions and impairment of notochord vac-uoles to defective axial elongation [52, 54]. While we did not observe these phenotypes in *wnt16*$^{-/-}$ mutants, we found that mutants exhibited reduced notochord cross-sectional area in concert with reduced notochord length. Because notochord volume is reduced, it is conceiv-able that notochord-derived Wnt16 influences notochord cell number and/or size, which in turn impacts axial elongation [53, 55]. While notochord is known to secrete signals that pat-tern somite, in our studies, *wnt16*$^{-/-}$ mutants did not exhibit missing somites or absence of the horizontal myoseptum, mutant phenotypes previously linked to absence of notochord differ-entiation [56]. Lastly, altered axial elongation has been observed in zebrafish mutants with pri-mary defects in notochord [52, 54] and muscle activity [57]. Moreover, myotome shape is regulated by mechanical coupling between future myotome and notochord [58]. These studies highlight that effects of loss of *wnt16* on notochord, spine, and muscle morphology may not be totally independent.

We identified a novel signaling domain in notochord sheath defined by expression of *wnt16*. Specifically, *wnt16* is expressed in the ventral midline, where it appears to contribute to chordacentrum width and sheath mineralization in the dorsal direction. To our knowledge this is the first study demonstrating the existence of morphogenetic signals at the ventral mid-line of the zebrafish notochord sheath. The presence of a signaling domain at the ventral mid-line of the notochord sheath provides a biological explanation for how chordacentrum mineralization initiates at the ventral midline in salmon [38], which appears to be similar in zebrafish. Our studies suggest that *wnt16* is primarily expressed in *col9a2*+ cells, which are linked to non-mineralizing domains [18]. It is possible that Wnt16 is secreted by *col9a2*+ cells where it acts as a morphogenetic cue that signals to *entpd5a*+ cells within mineralizing domains. Single-cell transcriptomic studies focused on notochord sheath populations may help determine whether the ventral midline of the notochord sheath is a source of morphoge-netic signals other than Wnt16. Finally, it is noteworthy that, in mammals, while most of the notochord disappears at sites of vertebral mineralization, it is believed that some notochord cells become entrapped within the primitive intervertebral disc, where they synthesize the

nucleus pulposus (NP) [59]. Previous transcriptome analyses of mouse notochord-derived cells found *Wnt16* to be the sole *Wnt* family member significantly upregulated during the notochord to NP transformation [60]. Thus, it is conceivable that the role of Wnt16 in the zebrafish notochord sheath has parallels to the role of WNT16 during the notochord-to-NP transition in mouse.

*wnt16* mutant skeletal phenotypes identified here differed with some previous studies. In our study, the primary skeletal phenotype of adult *wnt16* mutants—reduced centrum length and increased tissue mineral density—contrasts with that of Qu et al., who reported zebrafish homozygous for a presumptive *wnt16* null allele exhibit craniofacial abnormalities and loss of the caudal fin rays without a reduction of the spine length [13]. In our study, phenotypes in *wnt16*$^{-/-}$ mutant larvae were indistinguishable for three different alleles. Moreover, *wnt16*$^{-/-}$ mutants did not exhibit loss of caudal fin rays, in agreement with a *wnt16* presumptive null allele generated by McGowan et al. [14]. The allele of Qu et al. comprised a 11bp deletion in exon 3, which was generated by two gRNAs with different target sites. Differences in phenotypes in the studies of Qu et al. and those reported here could be due to different effects of mutant alleles on Wnt16 expression or function, off-target mutations, or differential activation of NMD-induced genetic compensation [34]. Developing a consensus around a *wnt16* null mutant skeletal phenotype will aid community efforts in developing zebrafish as a model for human skeletal genomic research [22, 23, 61–63].

Because *wnt16* mutants exhibit altered lean mass—the same trait that, in concert with BMD, is associated with variants at the *CPED1-WNT16* locus [5], our findings support the potential for WNT16 to act as a causal gene underlying bone and muscle pleiotropy at this locus. This notion is further supported by our reverse genetic screen, in which *wnt16* was a gene of major effect on lean tissue-related traits. Our studies support the hypothesis that pleiotropic variants at some loci act during embryonic or fetal development, where bone and muscle experience organogenesis through a highly interconnected gene network [5, 6]. This work also brings forth the possibility that the multiple independent GWAS signals at the *CPED1-WNT16* locus reflect different biological mechanisms by which *WNT16* contributes to musculoskeletal development, growth, and/or homeostasis, rather than different causal genes. Lastly, it has been speculated that genetic factors may determine peak muscle and bone mass in early life, which delays osteoporosis and sarcopenia in late life [64]. In this context, variants at the *CPED1-WNT16* locus originally identified to be associated with BMD in adults also influence bone mass in children [5, 65]. This, in concert with the fact that variants at the *CPED1-WNT16* locus are associated with pleiotropic effects on BMD and lean tissue mass in children, highlights that they operate within the musculoskeletal system early in life. Our study supports the hypothesis that causal variant(s) at the *CPED1-WNT16* locus alters the expression or function of *WNT16*, which regulates muscle and bone morphogenesis. This alters BMD and lean mass accrual in early life, which in turn contributes to peak BMD and lean mass later in life.

WNT16 is actively being investigated as a potential therapeutic target for osteoporosis [66, 67], raising questions regarding the potential role of *wnt16* in adult muscle. In zebrafish, in addition to dermomyotome, Pax7+ cells are additionally localized between myofibers, exhibiting an identical niche and proliferative behavior as mammalian satellite cells [19]. These satellite-like cells contribute to skeletal muscle repair in adult zebrafish and this process requires Pax7, similar to the mouse [68]. Previous studies have shown there are at least two populations of *pax7+* cells, defined by expression of *pax7a* and *pax7b*, that contribute to muscle repair [69]. In our scRNA-seq analyses, *wnt16* was expressed in both *pax7a+* and *pax7b+* positive cells, suggesting that wnt16+ cells do not overlap with a specific *pax7+* population. While we found no obvious changes in somitic *pax7a* expression in *wnt16*$^{-/-}$ mutant embryos, whether *wnt16* is necessary for activation of *pax7+* satellite cells following injury warrants future

consideration. In mouse, *Wnt16* is expressed in somites similar to zebrafish [70]. Moreover, *Wnt16* was a top differentially expressed gene in transcriptomic analyses of skeletal muscle in exercise-trained rats [71] and the most downregulated Wnt pathway component in the skeletal muscle of mice that have experienced spaceflight-induced muscle loss [72]. Thus, functions of Wnt16 in regulating zebrafish myogenesis could be conserved in mammals. Further studies are needed to understand the cellular mechanism by which *wnt16* regulates muscle mass, whether this regulation broadly exists in conditions of development, growth, and regeneration, and whether *wnt16* regulates muscle force production in addition to muscle size and shape. Such studies will help evaluate the potential therapeutic benefit of targeting WNT16 for fracture prevention through its influence on muscle, or for treating both tissues simultaneously.

Finally, our study raises questions regarding the evolutionary origins of bone and muscle pleiotropy at the *CPED1-WNT16* locus. In our studies, *wnt16*$^{-/-}$ mutants exhibited decreased fineness ratio, a measure of body shape in fishes that is correlated with swim performance [46]. Presumably, *wnt16* could contribute to a developmental module that facilitates bone and muscle morphological adaptions that confer increased fitness within different ecological environments. Interestingly, in humans, in addition to musculoskeletal traits, variants at the *CPED1-WNT16* locus are associated with anthropometric traits such as waist-hip ratio and BMI-adjusted waist-hip ratio [73]. Moreover, variants at the locus are associated with lumbar spine area (associated phenotype in the Musculoskeletal Knowledge Portal [74]), a measure of bone size. In fishes, it would be interesting to study whether genetic variations at the *wnt16* locus are associated with measures of bone and body shape. The identification of such associations would lend support to the concept that a better understanding of genetic determinants of musculoskeletal traits can be derived in part through an evolutionary understanding of the genome [75].

## Materials and methods

### Ethics statement

All studies were performed on an approved protocol (#4306–01) in accordance with the University of Washington Institutional Animal Care and Use Committee (IACUC).

### Animal care

Zebrafish were housed at a temperature of 28.5˚C on a 14:10 hour light:dark photoperiod. Studies were conducted in mixed sex wildtype (AB), mutant (*wnt16*$^{w1001}$, *wnt16*$^{w1008}$, *wnt16*$^{w1009}$, *cped1*$^{w1003}$) and transgenic (Tg(*ctsk*:DsRed), Tg(*sp7*:EGFP)) lines. Fish were housed in plastic tanks on a commercial recirculating aquaculture system and fed a commercial diet. Germline *wnt16*$^{w1001}$, *wnt16*$^{w1008}$, *wnt16*$^{w1009}$, and *cped1*$^{w1003}$ mutants were maintained through heterozygous inbreeding and clutches of mixed genotypes were housed together. For genotyping, PCR was performed using standard PCR conditions (35 cycles, 58˚C annealing temperature) with the following primers for *wnt16*$^{w1001}$ (F: 5'–CGGCTGCTCTGAT GACATCG– 3', R: 5'–TCCCAGCCTCACTGTTATGC– 3'); *wnt16*$^{w1008}$ (F: 5'–CATGCTCT CCGTGTCCTGTT– 3', R: 5'–ATCCTTGCGTCGCACCTTAC– 3'); *wnt16*$^{w1009}$ (F: 5'– CATGCTCTCCGTGTCCTGTT– 3', R: 5'–ATCCTTGCGTCGCACCTTAC– 3'); *cped1*$^{w1003}$ (F: 5'–CCTACAGGAGGCAGCCAATC– 3', R: 5'–CAGCGTAGAGACCCAAGCAG– 3'). Genotypes were identified by performing electrophoresis using high resolution (3%) agarose gels to resolve different sizes of PCR amplicons of wildtype and mutant alleles. Experimental animals were generated by incrossing heterozygous mutants except for generation of maternal-zygotic mutants which were generated by crossing homozygous *wnt16*$^{-/-}$ females with heterozygous *wnt16*$^{-/+}$ males. In some animals, the C-start response was invoked as described in [76].

## CRISPR-based gene editing

CRISPR mutagenesis was performed using the Alt-R CRISPR-Cas9 System from Integrated DNA Technologies (IDT). For each gene, gRNAs were generated by mixing the crRNA and tracrRNA in a 1:1 ratio, incubating at 95°C for 5 minutes and cooling on ice. Ribonucleoprotein complexes were generated by mixing the combined crRNA:tracrRNA gRNA in a 1:1 molar ratio with Alt-R S.p. Cas9 Nuclease (IDT) containing a 3XNLS sequence and incubated for 5–10 minutes at room temperature to produce the Cas9:gRNA RNP complex at a final concentration of ~25 μM for injection. RNPs were loaded into pre-pulled microcapillary needles (Tritech Research), calibrated, and 2 nL RNP complexes were injected into the yolk of 1- to 4-cell stage embryos. The cRNA guide target sequences used to generate germline mutants were as follows: *wnt16*$^{w1001}$ (5'—CTGCACTGTCAATAAAGCGG—3'), *wnt16*$^{w1008}$ and *wnt16*$^{w1009}$ (5' -AACC CGATACGCCATGACAG—3'), and *cped1*$^{w1003}$ (5'–GCGTAACTAGCTTTATCCTG– 3').

The reverse genetic screen of genes at the *CPED1-WNT16* locus was performed by generating somatic mutants using methods we have described previously [26]. The crRNA guide target sequences for the reverse genetic screen were as follows: *cped1* (GCGTAACTAGCTTTAT CCTG), *fam3c* (GTGAAGAACAACATTGGACG), *ing3* (CGATGGATCAGCTTGAGCAG), *tspan12* (GACGACAGGATGGACCACGG), and *wnt16* (CTGCACTGTCAATAAAGCGG). Somatic mutants were housed separately but at equal densities to uninjected, control AB clutchmates to maintain environmental consistency between somatic mutant and uninjected clutchmates [26]. For mutation efficiency analysis, somatic mutants were collected between 24–120 hpf, and genomic DNA was extracted. The CRISPR target region was amplified with the following primers using standard PCR conditions: *wnt16* (F: 5'–CGGCTGCTCTGATGA-CATCG– 3', R: 5'—TCCCAGCCTCACTGTTATGC– 3'), *cped1* (F: 5'–CCTACAGGAGG-CAGCCAATC– 3', R: 5'—CAGCGTAGAGACCCAAGCAG– 3'), *fam3c* (F: 5'—TTTCAGC CACCAGACCACTG– 3', R: 5'–ACCTCCAGTCCACATGTCAA– 3'), *tspan12* (F: 5'–AGA-GAGATCTTTAGCTTTGTCTTCT– 3', R: 5'–AGCAACGTCATTACCAGCGA– 3'), *ing3* (F: 5'–CCGGTTACATTTCCCACCAAAA– 3', R: 5'–CCAACCCAAAAACACTGATGGT– 3'). Sanger sequencing was performed with the appropriate PCR primer except for *fam3c*, which was sequenced with the following sequencing oligo: fam3c (SF1: 5'–AAGATCACCAGTG-GAGCAGC– 3'). CRISPR mutation efficiency analysis was performed using the Tracking of Indels by DEcomposition (TIDE) web tool [77]. Batch analyses were performed using the standard parameters for sequence decomposition analysis. From the "%WT" (percent wildtype, defined as the predicted frequency of zero indels) for each sample, we calculated the Mutation Efficiency, which we defined as 1—%WT.

## Myotome and notochord analysis

For brightfield imaging, embryos were collected at appropriate time points, dechorionated with fine forceps, anesthetized in 0.01% MS-222, and mounted in anesthetic in a deep well slide for imaging on a Zeiss Axio Imager M2 using a 5X objective. The fish were then euthanized and genomic DNA was extracted via incubation in proteinase K in tissue lysis buffer (10 mM Tris pH 8, 2 mM EDTA, 0.2% Triton X-100, 200 μg/ml Proteinase K) for 30 minutes at 55°C. The proteinase K was then deactivated by incubation at 90°C for 10 minutes, and the genomic DNA was used for genotyping to identify wildtypes, heterozygotes, and homozygote mutants. Image measurements were performed in Fiji [78].

## Myofibril analysis

For phalloidin imaging, 3 dpf embryos were collected, dechorionated with fine forceps, euthanized, and fixed in 4% paraformaldehyde/1X PBS overnight at 4°C. The heads were collected

for genomic DNA extraction and genotyping, and remaining tissues were stored in 1X PBS/ Tween20 (PBS-T) at 4˚C. Identified mutants and wildtype siblings were used for phalloidin staining, which was performed as follows: Larvae were permeabilized in 1X PBS/2% Triton X-100 for 90 minutes at room temperature, then incubated in 1:100 phalloidin/1:500 DAPI in 1X PBS/2% TritonX-100 overnight at 4˚C. Larvae were washed 3 x 5 minutes in PBS-T and mounted in 1.5% low-melting point agarose on a glass-bottom petri dish. Imaging was performed on a Nikon A1R confocal microscope with a Plan-Apochromat 20X/0.8 air objective, and image processing was performed in Fiji. All image stacks were rotated as needed in the x, y, and z planes to ensure the rostral-caudal axis of the fish was parallel to the x-axis using the TransformJ plugin [79]. The FibrilTool plugin was used for myofibril angle and anisotropy analysis [80], and the Stardist plugin was used for nuclei segmentation [81]. Myofibril angle was computed by averaging values measured in 4–5 consecutive epaxial myotomes located just posterior to the cloaca, from a single optical section within lateral fast muscle. Measurements were made such that an arrow that points upward has an angle of 90˚ and an arrow that points down has an angle of −90˚. For cross-sectional analysis, image stacks were resliced and compartments were segmented by manually tracing them using the polygon selection tool in Fiji. For each animal, measurements were computed by averaging values measured on two consecutive myotome segments just posterior to the cloaca.

## Vertebral mineralization analysis

At the appropriate time point, larvae were stained with a 0.2% calcein (Sigma) solution in fish water. A subset of fish were also stained with a 0.05% alizarin red (Sigma) solution in fish water 3 days prior to calcein staining. For imaging, stained zebrafish larvae were anesthetized in 0.01% MS-222 (Sigma) and mounted into borosilicate glass capillaries using 0.75% low melt-agarose (Bio-Rad) diluted in system water containing 0.01% MS-222. Capillaries were set on a custom 3-D printed holder to aid manipulation and rapid orientation of the specimen. Three-channel (GFP, DsRED, DAPI) images were collected on a high-content fluorescent microscopy system (Zeiss Axio Imager M2, constant exposure settings for all experiments) using a 2.5x objective (EC Plan-Neofluar 2.5x/0075). For each fish, a composite image stack (usually 3/1 images in the x/y directions; and optimized to 30-70 μm slice intervals in the z direction across the entire region of interest, usually about 9 slices; all at 2.58 μm/pixel) was acquired in mediolateral and anteroposterior views. Maximum intensity projections were generated from image stacks in Fiji for analysis. Following imaging, fish were collected for genomic DNA extraction and genotyping.

## RT-PCR

For RT-PCR analysis of the effects of mutant alleles on transcript expression and splicing, we designed primers that were located on the exons flanking the CRISPR-targeted exon. Total RNA was isolated from adult AB wildtype and homozygous mutant fish by homogenization in Trizol as per the manufacturer's protocol. cDNA synthesis was performed using the Superscript IV First Strand Synthesis System (Invitrogen), and 1μL of each cDNA was used for PCR using the DreamTaq Green DNA polymerase (ThermoFisher Scientific) in a 20μL volume reaction. Gel electrophoresis was performed on a single 2% agarose gel using 10μL of each PCR reaction, and bands were excised for gel extraction and Sanger sequencing to confirm the product. The following primers were used for PCR: *wnt16* (F: 5'–CGAGTGCCAAACT-CAGTTCA– 3', R: 5'- GCGTTGCTCTTTATCCTTGC– 3'), *cped1* (F: 5'–CTTTTGCCGGAG CGTTTCTG– 3', R: 5'–GGCTATCTGCCCATCGAAGT- 3'), *actb2* (F: 5'–GAAATTGCCG-CACTGGTTGT– 3', R: 5'–CGTAACCCTCGTAGATGGGC– 3').

## Transcriptomic analyses

For zebrafish single-cell transcriptome atlas analysis, data from the embryonic zebrafish single-cell atlas published in Farnsworth et al. [27] were downloaded from the Miller lab website (https://www.adammillerlab.com/resources-1). All analyses were performed using the Seurat package in R [82]. The somite cluster was subset, scaled using the ScaleData function, and subclustered. Briefly, clustering was performed using the FindClusters function using a resolution of 0.5. PCA, clustering, and UMAP analyses were performed using 20 dimensions. Cell cycle and *wnt* family member analyses were performed using the CellCycleScoring and VariableFeaturePlot functions, respectively. All results were visualized using Seurat or the dittoSeq package [83].

For RNA-seq analysis of notochord sheath cells, processed RNA-seq data from a study by Wopat et al., 2018 on notochord sheath cell populations isolated from 13 dpf Tg(*col9a2*: *GFPCaaX;entpd5a:pkRED*) zebrafish were downloaded from the NCBI GEO Database (GEO: GSE109176) [18]. The data contained count tables generated by the htseq-count tool [84] on the Galaxy Web platform for three FACS-sorted notochord sheath cell populations: the *entpd5a*+ cell population (n = 3), the *col9a2*+ cell population (n = 3), and the *entpd5a* +/*col9a2*+ cell population (n = 3). We uploaded the count data for the *entpd5a*+ samples and *col9a2*+ samples to the Galaxy Web platform, and used the public server at https://usegalaxy.org to perform differential gene expression analysis with DESeq2 using default parameters [85]. The results were plotted using Prism 8.4.3 (GraphPad Software, San Diego, CA).

## microCT scanning and analysis

For microCT imaging, animals were scanned at 90 dpf using a Scanco vivaCT 40 microCT scanner. Scans with 21 μm voxel size were acquired using the following settings: 55kVp, 145mA, 1024 samples, 500proj/180 ˚, 200 ms integration time. DICOM files of individual fish were generated using Scanco software, and maximum intensity projections of DICOM images were used for analysis. Two fish were scanned simultaneously in each acquisition. FishCuT analysis was performed as previously described [24].

For lean tissue volume calculations, DICOM files were opened in Fiji, and a slice from the approximate midpoint of the stack adjacent to the posterior swimbladder was selected. The lower and upper thresholds were automatically selected in Fiji using the Default and MaxEntropy threshold algorithms, respectively [78]. A custom MATLAB script was written to open DICOM files and segment voxels based on intensity into the following compartments: presumptive adipose (below lower threshold), lean (between lower and upper threshold), and bone (above upper threshold). For testing of segmentation accuracy, we randomly selected cross sections for each specimen, manually segmented lean tissue using the polygon selection tool in Fiji, and compared the area of lean tissue computed by our program against that computed manually. A similar process was performed for bone and presumptive adipose tissues. We used the lm function in R to perform a regression between manual- and program-generated areas.

For displaying soft tissue at high resolution, adult zebrafish samples were placed in 1% potassium iodide solution for 12 hours, placed in a custom-made sample holder maintaining hydration of the specimen [86], and scanned in a Skyscan 1271 (Bruker) at 3 μm voxel size using 60 kV, 140 μA, 4 frame averages, 3493 ms exposure time and a 0.5 mm Al filter. Reconstruction was performed using Nrecon (Bruker), whereby parameters of smoothing, ring artifact correction and beam hardening correction were kept constant for all samples.

## RNA *in situ* hybridization

RNA-ISH using RNAScope was performed as we have previously described [37]. Briefly, fixed tissues were washed in 1X PBS/0.05%Tween20 (PBS-T), then dehydrated in increasing graded concentrations of methanol and stored at −20˚C. Tissues were processed for embedding by first rehydrating to PBS-T, then cryoprotected by incubation in 15% sucrose/1X PBS, followed by 20% sucrose/1X PBS. Embedding was done in O.C.T. compound (Fisher Scientific) in Peel-A-Way embedding molds (Polysciences, Inc.). For sectioning, ∼15μm sections were made on a Leica CM1850 cryostat, collected onto charged slides, and stored at −20˚C. Probes were purchased for *wnt16*, *pax7a*, and *cdh11* from the manufacturer (ACD Bio). *In situ* hybridization was performed using the RNAScope 2.5 HD Duplex Detection kit essentially as per manufacturer instructions, with modifications described in [37]. Whole mount RNA *in situ* hybridization was performed as previously described [87]. The cDNA probes used were: *myog* [88], *ntla/tbxta* [89], and *pax7a* [90]. After staining, a small portion of the head was dissected for genotyping the *wnt16^w1001* allele. Representative embryos were mounted in 2.5% methyl cellulose for imaging.

## Statistical approach

For most results, data are reported from a single experiment. Each biological replicate represents one technical replicate. Empirical data are shown as either individual measurements or are reported as mean ± SEM. Group sizes (n) are reported in the figure panels themselves or in respective legends. Outliers were not identified; all data were included in statistical analyses. Multivariate analysis of vertebral data using the global test was performed using the globaltest package in R [24, 91]. All other statistical analyses were performed in GraphPad Prism. $p < 0.05$ was considered statistically significant in all cases.

## Supporting information

**S1 Fig. *wnt16* expression is differentially co-expressed with *pax7a* and *tbxta* in cells within the "somite" and "other" clusters, respectively.**
(TIF)

**S2 Fig. Isolation of *wnt16* mutant alleles.** (A) Sequence and genomic location of *w1001*, *w1008*, and *w1009*. Grey highlight indicates gRNA target sequence used for CRISPR-based gene editing with PAM underlined. (B) Predicted effects of alleles on amino acid sequence. (C) RT-PCR assessing *wnt16* transcript in *wnt16^w1001* mutants. No evidence of transcript reduction or alternative splicing is observed. (D) Calcein staining of 13 dpf animals show similar reductions in vertebral mineralization and post-cranial body length in *w1001*, *w1008*, and *w1009* mutants. (E) Quantification of mineralized area shows similar changes in mutants for all three alleles.
(TIF)

**S3 Fig. RNA ISH for markers of early muscle (*myog*, *pax7a*) and notochord (*ntla/tbxta*) differentiation in *wnt16^-/-* mutant embryos at 1 dpf.** Scale bar: 200 μm.
(TIF)

**S4 Fig. Zygotic and maternal zygotic *wnt16^-/-* mutant phenotypes.** P-values were determined using a two-way ANOVA with Fisher's LSD post hoc test. *$p < 0.05$, **$p < 0.01$, ***$p < 0.001$, ****$p < 0.0001$.
(TIF)

**S5 Fig. Heterozygous *wnt16*<sup>-/+</sup> mutants do not exhibit significant differences in bone measures compared to wildtype clutchmates.**
(TIF)

**S6 Fig. *wnt16*<sup>-/-</sup> mutants exhibit reduced standard length compared to wildtype and heterozygous clutchmates.** P-values were determined using one-way ANOVA with Fisher's LSD post hoc test. ***p<0.001, ns: not significant.
(TIF)

**S7 Fig. Analysis of *wnt16*<sup>-/-</sup> fish following allometric normalization for standard length.** *wnt16*<sup>-/-</sup> mutants exhibit significant differences for most measures (all except for Cent.Vol) when normalized for differences in standard length.
(TIF)

**S8 Fig. Correlation in tissue area computed using automatic and manual segmentation for presumptive adipose tissue (left), lean tissue (middle), and bone tissue (right).**
(TIF)

**S9 Fig. Inference of myomere morphology from vertebral measures.** (A) Schematic demonstrating correlated features. (B-D) *wnt16*<sup>-/-</sup> mutants exhibit altered centrum length and neural arch angle. P-values were determined using an unpaired t-test. **p<0.01, ***p<0.001.
(TIF)

**S10 Fig. *wnt16*<sup>-/-</sup> mutants do not exhibit obvious muscle pathology.** H&E-stained sections showing frontal section of the caudal region and corresponding magnified areas in 30 dpf animals (A) and transverse sections in adult animals (B). *wnt16*<sup>-/-</sup> mutants exhibited no obvious differences in muscle segmentation or fiber morphology. (C) Mutants and wildtype clutchmates exhibited full-length body flexions when the startle-induced C-start response was evoked.
(TIF)

**S11 Fig. Isolation of *cped*<sup>w1003</sup>.** (A) Genomic location of *w1003* and its corresponding location mapped to mouse *Cped1*. Locations of Cadherin and PC Esterase domains in mouse *Cped1* are from [49]. (B) Genomic sequence of *w1003*. Grey highlight indicates gRNA target sequence used for CRISPR-based gene editing with PAM underlined. (C) RT-PCR assessing *cped1* transcript in *cped1*<sup>w1003</sup> mutants. No evidence of transcript reduction or alternative splicing is observed.
(TIF)

**S12 Fig. Data used for calculating Z-scores for Fig 9D.**
(TIF)

**S1 Code. Sample MATLAB code used for computing lean tissue volume.**
(DOCX)

## Acknowledgments

The authors would like to thank Dr. Matthew Harris for sharing the Tg(*ctsk*:*DsRed*) line, Dr. Eric Thomas, Dr. Ivan Cruz, and Dr. David Raible for sharing reagents and technical advice, the ISCRM Aquatics Facility for animal care, and the Orthopaedics Science Laboratories for assistance with microCT scanning. The authors would also like to thank Dr. Keith Cheng, Dr. Chris Amemiya, members of the MSBL and Seattle Fish Club, and anonymous reviewers for helpful suggestions. One or more images in this manuscript were created with Biorender.com.

## Author Contributions

**Conceptualization:** Claire J. Watson, Lisa Maves, Bjorn Busse, Yi-Hsiang Hsu, Ronald Young Kwon.

**Formal analysis:** Claire J. Watson, W. Joyce Tang, Maria F. Rojas, Imke A. K. Fiedler, Andrea R. Cronrath, Lulu K. Callies, Ali R. Ahmed, Visali Sethuraman, Sumaya Addish, Adrian T. Monstad-Rios.

**Funding acquisition:** Claire J. Watson, Edith M. Gardiner, Bjorn Busse, Yi-Hsiang Hsu, Ronald Young Kwon.

**Investigation:** Claire J. Watson, W. Joyce Tang, Maria F. Rojas, Imke A. K. Fiedler, Ernesto Morfin Montes de Oca, Andrea R. Cronrath, Lulu K. Callies, Avery Angell Swearer, Ali R. Ahmed, Visali Sethuraman, Sumaya Addish, Gist H. Farr, III, Arianna Ericka Gómez, Jyoti Rai, Adrian T. Monstad-Rios.

**Methodology:** Claire J. Watson, W. Joyce Tang, Maria F. Rojas, Imke A. K. Fiedler, Ernesto Morfin Montes de Oca, Avery Angell Swearer, Ali R. Ahmed.

**Project administration:** Ronald Young Kwon.

**Software:** Ali R. Ahmed.

**Supervision:** Claire J. Watson, David Karasik, Lisa Maves, Bjorn Busse, Ronald Young Kwon.

**Validation:** W. Joyce Tang, Maria F. Rojas.

**Visualization:** Claire J. Watson, W. Joyce Tang, Gist H. Farr, III, Ronald Young Kwon.

**Writing – original draft:** Claire J. Watson, W. Joyce Tang, Imke A. K. Fiedler, Lisa Maves, Ronald Young Kwon.

**Writing – review & editing:** Claire J. Watson, W. Joyce Tang, Maria F. Rojas, Imke A. K. Fiedler, Ernesto Morfin Montes de Oca, Andrea R. Cronrath, Lulu K. Callies, Avery Angell Swearer, Ali R. Ahmed, Visali Sethuraman, Sumaya Addish, Gist H. Farr, III, Arianna Ericka Gómez, Jyoti Rai, Adrian T. Monstad-Rios, Edith M. Gardiner, David Karasik, Lisa Maves, Bjorn Busse, Yi-Hsiang Hsu, Ronald Young Kwon.

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
