## [Decision Letter · Decision Letter 0]

22 Sep 2022

Dear Dr Kwon,

Thank you very much for submitting your Research Article entitled 'wnt16 regulates spine and muscle morphogenesis through parallel signals from notochord and dermomyotome' to PLOS Genetics.

The manuscript was fully evaluated at the editorial level and by independent peer reviewers. The paper is nearly acceptable for publication pending the editorial changes as requested by one of the reviewers.

We therefore ask you to modify the manuscript according to the review recommendations. Your revisions should address the specific points made by each reviewer.

[LINK]

Yours sincerely,

FANXIN LONG

Academic Editor

PLOS Genetics

Gregory Barsh

Editor-in-Chief

PLOS Genetics

Reviewer's Responses to Questions

**Comments to the Authors:**

Reviewer #1: The authors have made substantial revisions addressing all previously raised concerns. The additional description of the mutants as well as details of methodology increases the rigor of the presented work. In addition, the added analysis of the role of wnt16 in notochord and spine patterning and development, significantly strengthens the paper. The authors now show a role of wnt16 in notochord patterning and recruitment of osteoblasts, which is in line with previous observations of impaired osteoblast recruitment during fracture healing in wnt16 mutants.

I would suggest some minor changes and additions to improve the readability and to clarify analytical procedures.

1. The section describing figure 4 (starting with ‘Next, we examined whether wnt16 contributes to specific myotome compartments or developmental axes’) is duplicated. I would suggest keeping the second version.

2. Figure 6, quantification of mineralized domain morphology. Please add how many fish and how many vertebrae per fish were analyzed in C-G. Which vertebrae were analyzed?

3. Figure 8B, it is unclear what ‘a’ refers to. Is this the length or the posterior height of the fish?

4. Figure 9A, the gene names are unreadable.

Reviewer #2: The authors have thoroughly addressed the issues raised by the reviewers and included much more quantitative analysis of the phenotypes associated with the loss of wnt16 function during zebrafish musculoskeletal development.

Reviewer #3: The authors have made substantial revisions both to the paper and to the data which is included. These additions greatly strengthen the paper. They have addressed all of my previous comments and I have no further suggestions.

**Have all data underlying the figures and results presented in the manuscript been provided?**

Reviewer #1: Yes

Reviewer #2: Yes

Reviewer #3: Yes

PLOS authors have the option to publish the peer review history of their article (what does this mean?). If published, this will include your full peer review and any attached files.

Reviewer #1: No

Reviewer #2: No

Reviewer #3: **Yes: **Chrissy Hammond

---

## [Editor Report · Decision Letter 1]

24 Oct 2022

Dear Dr Kwon,

We are pleased to inform you that your manuscript entitled "wnt16 regulates spine and muscle morphogenesis through parallel signals from notochord and dermomyotome" has been editorially accepted for publication in PLOS Genetics. Congratulations!

Yours sincerely,

FANXIN LONG

Academic Editor

PLOS Genetics

Gregory Barsh

Editor-in-Chief

PLOS Genetics

Comments from the reviewers (if applicable):

**Data Deposition**

http://datadryad.org/submit?journalID=pgenetics&manu=PGENETICS-D-22-00991R1

**Press Queries**

---

## [Editor Report · Acceptance letter]

2 Nov 2022

PGENETICS-D-22-00991R1 

wnt16 regulates spine and muscle morphogenesis through parallel signals from notochord and dermomyotome 

Dear Dr Kwon, 

We are pleased to inform you that your manuscript entitled "wnt16 regulates spine and muscle morphogenesis through parallel signals from notochord and dermomyotome" has been formally accepted for publication in PLOS Genetics! Your manuscript is now with our production department and you will be notified of the publication date in due course.

With kind regards,

Zsofia Freund

PLOS Genetics

On behalf of:
